



# Monitoring offshore wind farm power performance with SCADA data and advanced wake model

Niko Mittelmeier[1],Tomas Blodau[1], Martin Kühn[2]

[1]Senvion GmbH, Überseering 10, 22297 Hamburg, Germany
[2]ForWind – Carl von Ossietzky University of Oldenburg, Ammerländer Heerstraße 136, 26129 Oldenburg Germany

*Correspondence to*: Niko Mittelmeier (niko.mittelmeier@senvion.com)

**Abstract.** Wind farm underperformance can lead to significant losses in revenues. Efficient detection of wind turbines operating below their expected power output and immediate corrections help  maximise asset value. The presented method estimates the environmental conditions from turbine states and uses pre-calculated power matrices from a numeric wake

model to predict the expected power output. Deviations between the expected and the measured power output are an indication of underperformance. The confidence of detected underperformance is estimated by detailed analysis of uncertainties of the method. Power normalisation with reference turbines and averaging several measurement devices can reduce uncertainties for estimating the expected power. A demonstration of the method's ability to detect underperformance in the form of degradation and curtailment is given. Underperformance of 8% could be detected in a triple wake condition.

**1. Introduction**

To increase confidence in offshore wind energy investments, investors need reliable technical solutions. The two pillars of system reliability are operational availability and the ability to achieve predicted power performance. In the wind industry, the common standard availability definition (IEC 61400-26-1, 2011) defines  that the system is ready to operate. However within this standard there is no indication about the quality of the power performance of the whole wind farm under given

conditions. The key to an economic investment is a function of quality and quantity.

Quantity is linked to availability and wind turbines can provide much SCADA (supervisory control and data acquisition) information which enables analysis of the time based (IEC 61400-26-1, 2011) and production based availability (IEC TS 61400-26-2, 2014).

A power curve verification test according to the international standard (IEC 61400-12-1, 2005) proves the quality of the power performance of a single turbine in specific conditions using a hub height met mast typically for a very limited period of time. Work on a second edition of the IEC is ongoing which would allow smaller masts in combination with a remote sensing device (e.g. LiDAR or SODAR) and is expected for release by the end of 2016. For most turbines in a typical offshore wind farm, this verification of the performance is not suitable due to wake effects. And the installation and





maintenance for an offshore met mast is very expensive. Quantifying changes in power production based on wind speed measurements from nacelle anemometry relies on the quality of the device itself and its transfer function which should accounts for the flow distortion behind the rotor. Using this approach leads to an increase in uncertainties. (IEC TS 61400-26-2, 2014)

Efficient detection of underperformance of wind turbines increases asset value (Albers, 2004). Incorrect parameter settings, degradation of the blades, pitch or yaw errors all lead to less production than expected. We differentiate between degradation and curtailments. A turbine that is degraded reaches rated power, but does not fulfil its expected power curve. A curtailed turbine has a limited power output which has been externally applied and is below its expected power.

Several approaches to evaluate the performance of a whole wind farm have been published before. Upwind turbines influencing the free flow for downwind turbines, called wake effects, add complexity to the task. Much work has been done by Axel Albers (Albers and Gerdes, 1999), (Albers et al., 2002) and (Albers, 2004) who has investigated performance verification and underperformance detection methods based on correlations between the individual turbines and the wind

farm as a whole. Albers has also looked into the possibility of verifying wind farm performance by comparison with wake flow models. But at that time he concluded that the models have to be further development especially for complex terrain.

An international standard (TC88 WG6, 2005), which proposed to use a met mast and a wake model for performance verification of the wind farm as a whole, could not be established due to high uncertainties and inaccurate results and the

support from the Technical Committee members crumbled.

Further investigations are necessary to obtain a reliable and automated method, to detect underperformance or curtailment at individual turbines in a wind farm. (Mittelmeier et al., 2013) presented a new method to compare expected power results generated from complex wake models (pre-calculated and stored in matrices) with the actual wind turbine power output to

detect underperformance in multiple wake situations. To lower uncertainties the method is based on ratios between the observed turbine and all reference turbines in the wind farm. This method relies on measurements from a met mast to determine the environmental conditions. Especially for offshore sites, a met mast is very expensive and therefore often not available. Furthermore, with increasing size of wind farms, the assumptions of one measurement position being representative for the whole offshore wind farm is not valid (Dörenkämper, 2015).

The purpose of this paper is to present the results of extending this wind farm performance monitoring method (Mittelmeier et al., 2013) by using SCADA instead of met mast data. A new combination of methods to obtain representative environmental condition and further optimisation potential for wake models fine-tuned by SCADA data is presented and an





estimation of the uncertainty of these methods is given. Hence the presented method in (Mittelmeier et al., 2013) will be available with no requirement for installation of measurement equipment such as met masts or LiDAR.

In Section 2 the general approach of the method (Mittelmeier et al., 2013) is recalled. A new approach to generate a virtual

met mast from SCADA data is explained in detail in Section 2.1. The wake model optimisations are described in Section 2.2. A closer look at the uncertainties of the method especially in relation to the establishment of a virtual met mast is discussed in Section 2.3. In Sections 3, 4 and 5 results for a demonstration case are presented, followed by a detailed discussion and the final conclusions.

## 2. Methods

To detect underperformance of a wind turbine, we estimate the expected turbine power $P_\pi$ (predicted power) with a wake model for the actual condition and compare its result with the actual measured power $P_\mu$. A deviation, higher than a certain threshold indicates underperformance.

The accuracy and calculation speed of a wake model are dependent on the degree of simplifications that are made to describe

the real wind farm flow. Fewer simplifications will increase computation time and accuracy to a certain degree and therefore improve the underperformance detection capabilities, on the other hand, a performance monitoring method needs to predict the power $P_\pi$ in real time or faster.

To be able to use a more sophisticated and more computational expensive wake model on a common personal computer, the power output $P_{\pi,i,j}$ can be pre-calculated for each wind turbine for each wind speed bin $i$ and wind direction bin $j$ and saved

in a two dimensional matrix. The predicted power output $P_\pi$ is derived from the matrix with linear interpolation knowing the measured wind speed and wind direction.

Additional information about the turbulence intensity, pressure, temperature and humidity from additional devices could be used to increase the dimensions of the power matrix and may add accuracy. As we are focusing on a monitoring method that

uses only SCADA data, we will discuss and demonstrate one way to extract an useful wind speed and wind direction for this purpose in Section 2.1.

Commonly used power measurements are averages of 10 minutes. Due to the fact that there is a high scatter on power measurements for the same wind speed and wind direction bin, averaging $N$ quantities of 10 minutes time samples is

necessary until the power value converges to a satisfactory degree. The power matrix and the averaging $N$ are derived in a pre-process as shown in Figure 1, which gives an overview on the whole performance monitoring process.





The power of the wind turbine under observation $P_{ob}$ is correlated to the power of a reference wind turbine $P_{\text{ref}}$. This leads to a normalized power curve with much lower slope in a wide range of partial load (See Figure 2) and therefore decreases sensitivity on wind speed measurement uncertainty.

This is described in Eq. (1) and Eq. (2).

$$\mu = \frac{1}{N}\sum_{n=1}^{N}\frac{P_{\mu ob_n}}{P_{\mu ref_n}},$$  (1)

$$\pi = \frac{1}{N}\sum_{n=1}^{N}\frac{P_{\pi ob_n}}{P_{\pi ref_n}},$$  (2)

10   where $P_{\mu ob}$ and $P_{\pi ob}$ are the measured and predicted power of the observed turbine. $P_{\mu ref}$ and $P_{\pi ref}$ are the measured and predicted power of the reference turbine.

The underperformance indicator can be described with Eq. (3):

$$\eta_{ob,ref} = 100\,\% \cdot \left(1 - \frac{\pi}{\mu}\right),$$  (3)

where $\eta_{ob,ref}$ describes the deviation of the measured correlation and the model predicted correlation in percent valid for the selected turbine pair of one reference ($ref$) and one observed ($ob$) turbine. Having the measured power correlation in the nominator increases the sensitivity. The underperformance interval range of the indicator is in this way between $[0, -\infty[$ . Non-operating turbine values have to be filtered out.

If $\eta_{ob,ref}$ is larger than the uncertainty (Section 2.3), underperformance has been detected. This correlation is repeated for each combination of turbines which leads to $n \cdot (n-1)$ results (n = number of turbines in the farm). This adds further confidence to the detection, because an underperforming turbine will meet the criteria several times.

**2.1 Determination of environmental conditions**

25   **2.1.1 Wind direction**

The first step is to derive a wind direction $\vartheta$ for each 10 min interval. Before doing so, the wind vane alignment must be checked by comparing one turbine's vane to the direction of the wake deficit on a downstream turbine. All other wind vanes in the farm are then referenced to this vane and any bias is corrected with the mean difference between the reference vane and the corrected vane. After applying this correction, the wind direction from all wind vanes are averaged in the complex





area to account for the wind direction discontinuity at the beginning/end of the scale, after removing outliers, defined by the $\pm 1{,}5 \cdot$ IQR (interquartile range).

### 2.1.2 Wind speed

Having determined an averaged wind direction we are now able to derive the averaged free flow wind speed. For this task
we use the nacelle anemometry but only from wind turbines that are not affected by upwind turbines. To determine whether a turbine is affected by an upwind turbine or not we use the specification for power curve measurements from the international standard (IEC 61400-12-1, 2005). Each turbine location is checked against all other turbine locations according to the averaged wind direction. This is done within a Cartesian coordinate system were $x$ represents the easting and $y$ being the northing. The wind turbine of interest $WT_i$ is located at the position $(x, y)$ and the turbine wake is from the turbine $WT_0$
at location $(x_0, y_0)$.

$$\alpha = 1{,}3 \cdot \arctan\left(2{,}5 * \frac{D_n}{L_n} + 0{,}15\right) + 10 \,, \tag{4}$$

Equation (4) is proposed by (IEC 61400-12-1, 2005) and describes the width of the wake angle seen by the downwind
turbine, where $D_n$ is the rotor diameter of the upwind turbine and $L_n$ the distance between the two turbines described by Eq. (5):

$$d_x = |x - x_0| \,,$$
$$d_y = |y - y_0| \,,$$
$$L_n = \sqrt{d_x^2 + d_y^2} \,, \tag{5}$$

With Eq. (6) the orientation $\beta$ can be derived (See Figure 3). $\beta$ describes the angle between the wake inducing turbine and the northing.

$$\beta = \begin{cases} \frac{\pi}{2} - \arctan\left(\frac{d_y}{d_x}\right) & x_0 > x \text{ und } y_0 > y \\ \frac{\pi}{2} + \arctan\left(\frac{d_y}{d_x}\right) & x_0 > x \text{ und } y_0 \leq y \\ 0 & x_0 = x \text{ und } y_0 > y \\ \pi & x_0 = x \text{ und } y_0 \leq y \\ \frac{3}{2}\pi - \arctan\left(\frac{d_y}{d_x}\right) & x_0 < x \text{ und } y_0 \leq y \\ \frac{3}{2}\pi + \arctan\left(\frac{d_y}{d_x}\right) & x_0 < x \text{ und } y_0 > y \end{cases} \,, \tag{6}$$





Each quadrant and the north inconsistency need different conditions. With $\beta$ and the wind direction $\vartheta$ the turbine wake indicator $\gamma$ can be described as:

$$\gamma = \begin{cases} |\beta + 360 - \vartheta| - \frac{\alpha}{2} & 0 < \beta < 90 \text{ und } 270 < \vartheta < 360 \\ |\beta - 360 - \vartheta| - \frac{\alpha}{2} & 270 < \beta < 360 \text{ und } 0 \leq \vartheta < 90 \, , \\ |\beta - \vartheta| - \frac{\alpha}{2} & \text{else} \end{cases} \qquad (7)$$

The wind turbine of interest $WT_i$ is categorized as waked turbine for $\gamma < 0$. The wind speed for the virtual met mast is therefore the average of the subset of the nacelle anemometer signals from all wind turbines with $\gamma > 0$.

### 2.2 The wake model

The wake model is a key factor in our performance monitoring method. Several benchmark tests have been published with a
10 large variety of different models (Gaumond et al., 2012), (Réthoré et al., 2013) and (Steinfeld et al., 2015). And research is still ongoing to further improve prediction accuracy of such models.

In Figure 1 we highlight that the wake model and its tuning is part of the pre-process. The performance monitoring method itself is based on linear interpolation from the result matrices only. In (Mittelmeier et al., 2015), we have identified three key
parameters for the tuning of the wake model (stability, wind direction uncertainty and wake drift). Figure 2 gives an example of how the different key parameters change the wake model results. The left plot visualises the active power of a turbine in wake normalised with a free flow condition in 6.3 D distance. The 0° on the *x*-axis locates the full wake situation according to the simulation. The right plot is a representation of the same data as normalised power curve with wind speed on the *x*-axis normalised with the wind speed, when wake effects fade away due to pitching activities of the upwind turbine.

In the first step, the wake model needs to be set up with the right atmospheric stability parameters. An increasing stability will cause higher wake losses and therefore shift the wake plot vertically down (from red rhombus to black triangles).
The next two steps are applied on the wake model results which need to be calculated for a directional resolution of 0.5° and for each wind speed bin of 1 m/s. This resolution was proposed by Gaumond et al.(Gaumond et al., 2014) for his method to
25 account for measurement uncertainties related to the wind direction which is the second key parameter in our tuning process. In his paper, three main sources of uncertainty are mentioned: The yaw misalignment of the reference turbine, the spatial variability of the wind direction within the wind farm and the variability of wind direction within the averaging of a 10 min interval. This causes a higher scatter in the data and leads to averaging effects that are not modelled in the simulation. In a post process each wind direction is averaged with weighted neighbouring results. A Gaussian distribution with a standard
deviation $\sigma_a$ has been proposed as a weighting function. The effect of this step is visualised in Figure 2 (red rhombus are



without and orange points are with $\sigma_a$ weighted averaging). In (Mittelmeier et al., 2015), we could show, that $\sigma_a$ is a function of the wind speed, decreasing with higher wind speeds.

Looking at the full wind rose for an AEP estimation, the Gaussian averaging has no impact on the result (Gaumond et al., 2014). But the smaller the wind direction bin size, the larger the prediction error made by the wake model. Hence it is crucial

for our monitoring method to increase accuracy for smaller wind direction bin sizes which will decrease the uncertainty of the method.

The third tuning parameter is correcting the bias of the wind direction in the wake model results (Wake drift). Large Eddy simulations made by Vollmer et al. [22] confirm this behaviour. In the following section, we will have a closer look at the uncertainties of the proposed method.

**2.3 Uncertainties and underperformance criteria**

It is essential to understand the uncertainties of the method to judge the confidence in underperformance detection. Any false alarm can cause unnecessary trouble shooting.

For this evaluation, we follow the "Guide to the expression of Uncertainties in Measurements" (GUM) (JCGM, 2008), which distinguishes between statistical Type A and instrumental Type B uncertainties. The important measurands of the method are

the measured power and the predicted power for each wind turbine under observation and for reference $(P_{\mu ob}, P_{\mu ref}, P_{\pi ob}, P_{\pi ref})$. For the measured power $P_\mu$ we only use Type B, because each measurand is obtained from different environmental conditions and therefore statistical Type A uncertainties are not applicable. The combined uncertainty can be derived with Eq. (8).

$$u_c(P) = \sum_{k=1}^{K}(c(P)_k \cdot u_k)^2 \ , \tag{8}$$

where $c_k$ is the sensitivity factor and $u_k$ the uncertainty of the k-th component of the measurement chain. For the predicted power $P_\pi$, we are using a combined uncertainty with statistical type A uncertainties, being the experimental standard deviation of the mean from the wake model predictions and type B uncertainties which conclude from the instrument devices

to estimate wind speed and wind direction. Table 1 shows the uncertainty components of the predicted power $P_\pi$ and provides the sensitivity factors, with $P_{\pi i,j}$ being the power value in the matrix referring to the wind speed bin $i$ and the wind direction bin $j$. $V_{i,j}$ being the wind speed and $\vartheta_{i,j}$ being the wind direction of the element. In Table 2, the corresponding components for the uncertainty of the measured power $P_\mu$ are listed.

The authors experience in power curve verifications has shown that the combined uncertainty of the measurement chain that includes a met mast and all its devices is usually about 4 % to 6 %. In our case, the wake model will add further uncertainties which would lead to even higher values and therefore yields an inacceptable rate for underperformance detection. To lower



this impact, the monitoring method is based on normalised measurements and normalised predictions. An error at the estimated wind speed has a much lower impact on the ratio of the power of two turbines than on their absolute power performance. The uncertainty for Eq. (1) can be described as:

$$u(\mu) = u\left(\frac{P_{\mu ob}}{P_{\mu ref}}\right) =$$

$$= \frac{P_{\mu ref}}{P_{\mu ob}} \cdot \sqrt{\left(\frac{u_c(P_{\mu ob})}{P_{\mu ob}}\right)^2 + \left(\frac{u_c(P_{\mu ref})}{P_{\mu ref}}\right)^2}, \tag{9}$$

Equation (9) explains how to calculate the uncertainty of a summation in quadrature for division (Bell, 2001). The equation is equivalent for $u(\pi)$ and is applied on each 10 min sample.

With the two uncertainties $u(\mu)$ and $u(\pi)$ being independent the standard propagation of errors for $\eta$ can be simplified according to (Ku, 1966) to the following equation:

$$u^2(\eta) = \left(\frac{\partial \eta}{\partial \mu}\right)^2 u^2(\mu) + \left(\frac{\partial \eta}{\partial \pi}\right)^2 u^2(\pi), \tag{10}$$

which leads to an uncertainty in $\eta$ of:

$$u(\eta) = \frac{100}{\mu} \sqrt{u^2(\pi) + \left(\frac{\pi}{\mu}\right)^2 u^2(\mu)}, \tag{11}$$

The uncertainty derived by Eq. (11) is around 7 % and can be displayed as a bandwidth around the underperformance

indicator $\eta$, visualized in Figure 4. The confidence level is one standard deviation, which is considered to be acceptable for underperformance detection.

In the next step we need to estimate the required number of power samples $N$ for averaging (see Eq. (1) and Eq. (2)). This is directly linked with the earliest point in time when underperformance can be detected. We define this point as having a lower prediction error (with the optimal turbine operation model) than the prediction error derived by the model with the erroneous

data taking the uncertainty into account.

## 3 Results and Demonstration

The objective of this paper was to present a developed method, using only SCADA data and pre-calculated numerical wake model results to detect underperformance at wind turbines in waked conditions within the wind farm. We have chosen the



Ormonde wind farm to demonstrate the new method. The 30 turbines have a rated power of 5 MW and are owned by Vattenfall. The wind farm is located in the Irish Sea 10 km west of the Isle of Walney.

The farm layout displayed in Figure 5 is structured in a regular array which allows comparison of several single wake, double wake and triple wake situations. The turbine distance for the investigated wake situation is 6.3 D. The neighbouring
rows are at 4.3 D. To simplify the demonstration of underperformance detection, we selected four turbines in one row. With south westerly wind direction, we focused on single wake, double wake and triple wake conditions behind turbine number 26 for a sector of 30° around the full wake situation. Two years of SCADA data were used for the following demonstration.

### 3.1 Environmental condition of demonstration wind farm

### 3.1.1 Wind direction of demonstration wind farm

The quality of the derived wind direction is visualized by plotting a histogram (Figure 6) for the full data set of two years with each count being the differences between a single wind vane measurement and the corresponding mean wind direction for the averaged period. The deviation of the single wind vanes from the averaged wind direction is nicely described by a Gaussian distribution with standard deviation of 3.1°. This value is usedfor the uncertainty of the wind direction Table 1 is referring to.

### 3.1.2 Wind speed of demonstration wind farm

Figure 7 demonstrates the quality of the virtual met mast derived with the equations from Section 2.1.1 and Section 2.1.2. The average wind speed of all nacelle anemometers has been normalised by the averaged nacelle anemometer wind speed of the wake free subset. The full data is binned into 2° and plotted against the averaged wind direction. The errors bars indicate the experimental standard deviation of the mean (JCGM, 2008). We obtain a quite good agreement with the Fuga model
which has been used with a Gaussian averaging of standard deviation equal to 4°. So far there is no instruction available on how to determine this standard deviation which should take wind direction uncertainty into account (Gaumond et al., 2014). We have chosen this value, because of a quite nice fit with the SCADA data. A linear regression between the standard model results (red dashed line) and the SCADA measurements equals $R^2 = 0.975$. The improved model (green solid line) gives an $R^2 = 0.985$.

When considering the demonstration sector of 30° around the full wake alignment behind wind turbine 26, the free flow wind speed can also be described by a Gaussian distribution (Figure 8) with a standard deviation of 0.46 m/s.

This information is important for the investigation of the uncertainties Table 1 is referring to.





### 3.2 Wake model for demonstration wind farm

For the demonstration of the described method, we used the Fuga wake model which uses linearized Reynolds Averaged Navier Stokes equations developed by (Ott et al., 2011). With the second version of the Software new features were added (Ott and Nielsen, 2014) to account for different atmospheric stabilities and for wind direction uncertainties. The results for

this paper have been produced with Fuga (version 2.8.4.1). We have chosen this wake model for two reasons: Firstly, there is already a confident number of validations with measurements published (Gaumond et al., 2012), (Mortensen et al., 2013) and (Steinfeld et al., 2015) and secondly, the Gaussian averaging feature described by Gaumond et al. (Gaumond et al., 2014) is already implemented.

To get a more reliable monitoring model we need to calibrate the wake model settings and compare several different calculation results with measured SCADA data.

We identified three steps to obtain a better match between the power modelled by the wake model Fuga and the measurements. Firstly, the standard deviation $\sigma_a$ to account for the wind direction uncertainty was found to be decreasing with increasing wind speed. Secondly, the parameter to model the effect of atmospheric stability $\zeta_0$ was set to more stable

conditions and thirdly, the centre of the wake was found to be shifting towards starboard when traveling downwind. Approximately 2.5° in the single wake and an additional 1° is added with every turbine adding an additional wake to the flow. This results in a total offset of 4.5° for the triple wake referenced to the global wind direction. One possible explanation for this behaviour is the fact, that the upwards moving blade diverts the flow with higher wind speeds downwards to regions with lower wind speeds and the downwards moving blade causes the opposite. This results in a higher

wind speed on the port side than on the starboard side of the wake and leads to a drift of the wake centre. A second explanation can be derived from the Coriolis force, which leads to an increased force to starboard on accelerating air particles. Large Eddy simulations made by Vollmer et al. confirm this behaviour (Vollmer et al., 2014).

Figure 5 shows the layout of the demonstration wind farm. The plot indicates the estimated wakes with heat colours. The

row of turbines behind turbine 26 has been selected for the validation of the wake model settings. The benchmark are simulations for neutral conditions with none of the post processing's mentioned in section 2.2 to take wind direction uncertainty, atmospheric stability and wake shifts into account. Figure 9 demonstrates the improvement of model prediction and its capabilities for single wake, double wake and triple wake situation. The left column visualises wake deficit plots where the power has been normalized with the free flow turbine, as function of the wind direction, centred to the full wake.

The data is filtered for wind speed of 8±1 m/s. The right column are normalized wake power curves. The power, normalized with free flow power, is shown as function of the wind speed, normalized with wind speed at rated power for the waked turbine. This data has been filtered for a wind direction sector of 5°. The optimised simulation results with the green diamonds follows the SCADA data with the black dots much closer than the benchmark case marked as red triangles. The





error bars indicate one standard deviation of the measured SCADA data at each bin. The three fine-tuning steps decreased the prediction error in a full wake with ±5° sector width from 7% to 1.5% (Mittelmeier et al., 2015).

Having now an optimised wake model, the first two steps of the pre-process (Figure 1) are accomplished and the matrices for
the "predicted power" can be established. In the next Section, the detection of underperformance has been demonstrated with real data and two test cases.

### 3.3 Demonstration Case

Two years of SCADA data have been contaminated with two different error types. The first manipulation simulates a degradation of 8 % for which the original data set that has been used to calibrate the model, is multiplied by 0.92. According
to the findings in Section 2.3 a degradation of 8 % is just high enough to distinguish from the uncertainties of a turbine in triple wake. The second test case is a simple power curve curtailment at 60 % rated power.

In Figure 10 the normalized power as function of the normalized wind speed in displayed in a scatterplot. The coloured points in green represent correct turbine performance (P_optimal). The yellow dots (P_degraded) describe the degradation
and the red dots (P_curtailed) are the data with the curtailment. Measurements from above rated wind speed are removed to concentrate on the part of the power curve where underperformance is more difficult to detect.
Working with relative predictions the prediction error in percent rises with lower power production at low wind speeds. Therefore the model is supposed to treat only measurements above 5 m/s.

To further increase the certainty of the result, we calculate the underperformance indicator for each turbine with any other possible combination of reference turbine. For the whole wind farm of 30 turbines this would lead to 870 combinations. For simplification in this demonstration, we are only focussing on the four turbines in the row behind turbine 26.

At first, we need to estimate the required number of power samples $N$ for averaging. This is directly linked with the earliest
point in time when underperformance can be detected. We define this point by having a lower prediction error with the optimal turbine operation model than the prediction error derived by the model with the erroneous data taking the uncertainty into account. This can be visualized in Figure 11. The graphs present the accumulated level of underperformance $\eta$ as function of the number of samples ($N$ values). The data with the error appears in a solid line and grey uncertainty margin. The dashed green line represents the data with the turbine in optimal operation. The lowest quantity $N$, where the "optimal"
(green dashed) line confirms a lower $\eta$ than the border of the grey area, is the point where we highlight underperformance with sufficient certainty. Figure 11 demonstrates the two test cases with a turbine under curtailment at different wake situations and the corresponding situation for a degraded turbine. A wake model bias has been corrected in such a way, that the results for the optimal turbine prediction with the full two years data equals to zero.



In Table 3 we have listed the $N$ values which are necessary to measure for each case until underperformance can be detected with sufficient certainty. They can be translated into hours by $N/6$ as we are using 10 min averages.

Figure 12 is a graphical representation of the development of uncertainty with increasing number of averaging data. The free
flow has a comparatively low uncertainty in comparison with the three wake situations. All four graphs have higher uncertainty in the beginning and quickly decrease with increasing $N$. At approximately $N = 150$ the uncertainty of all three wake states has dropped at least once below 8 %. With $150 < N < 500$, $u(\eta)$ is still very unstable and stretches between 8 % and 10 %.
A clear additional drop, even below 7 %, can be seen from $500 < N < 1000$. Beyond $N > 1000$, $u(\eta)$ stabilizes
towards a more and more horizontal graph. In Table 4 the corresponding uncertainty for the estimated first time of detection is listed.
The power curve scatter plot of all four wake conditions with the number of quantities, necessary for detection are visualized in Figure 13.

**4. Discussion**

The model was able to detect the selected demonstration error cases after a certain averaging time. With the proposed sources of uncertainty and the described method to obtain a combined level a very clear increase of uncertainty can be seen from free flow to wake condition cases. The reason for this behaviour can be led back to the normalization procedure. The largest source of uncertainty is usually the wind speed measurement, followed by the wind direction measurement (category B uncertainties). Looking at the sensitivity factor for both readings, which are based on the slope of the quotient between
neighbouring normalized matrix cells, they approximately equal to zero for the free flow case. Therefore only category A uncertainties are left, which quickly decrease with increasing number of measured values N.

In our example, the curtailment took less than 130 values to be detected (see Table 3). This of course is very dependent on the wind distribution. The wind has to be high enough to force the turbine into the underperformance. At rated wind,
detection is much faster than at wind varying around the power limitation. The right column in Table 3 is showing the total values $N$ and in brackets the values considering only wind speeds high enough to force the turbine into the curtailment. The $N$ values to the first detection in Table 3 increase with each additional wake added to the flow. Furthermore, the figures show that curtailments (values in brackets) can be detected earlier than degradation.

The tuning of the wake model is an essential part of the method. The key tuning parameters have been estimated by trying to obtain the best fit with the SCADA data. This is a clear weak point of the method and further investigations are necessary to find ways to predict the right settings without measured data. Without such tuning, each of these parameters will contribute



as an additional source of uncertainty and therefore reduce the accuracy. A further improvement could be to extend the dimensions of the look up tables (matrices) with atmospheric stability. (Dörenkämper et al., 2012) could show that the influence on the development of wind turbine wakes is measurable. A link from SCADA data to atmospheric stability would be needed. An investigation is planned for future work.

The sensitivity of the underperformance indicator $\eta$ states the measured power correlation in the nominator of Eq. (3). In this way the interval range increases from $[0, -100]$ to $[0, -\infty[$ . Division with 0 is prevented by filtering non-operating conditions. The increased interval leads to a higher sensitivity and therefore further reduces the $N$ values for the first underperformance detection.

Using wind speed and wind direction measurements derived from a large number of devices can lead to acceptable levels of uncertainties although each single device for itself has comparably high uncertainties as described in more detail in the power verification standard using nacelle anemometry (IEC 61400-12-2, 2013).

### 5. Conclusion

A method for offshore wind farm power performance monitoring with SCADA data and advanced wake models was introduced. Wind speed and wind direction have been extracted from all devices in the wind farm to obtain a global measurement for the whole wind farm. In this way, the level of uncertainty could be lowered compared to a single nacelle measurement. Furthermore the uncertainties in performance level prediction could be reduced by normalization and referencing correlations. A suitable wake model was chosen, calibrated with SCADA data and used in a demonstration case.

Here the method was capable of detecting a degradation of 8 % with the confidence of one standard deviation. The described method can be used after a wake model recalibration with approximately two years of wind farm SCADA data. This would enable a real time monitoring from then on for the rest of the operational lifetime.

### Acknowledgement

The presented work is partly funded by the Commission of the European Communities, Research Directorate-General within
the scope of the project "ClusterDesign" (Project No. 283145 (FP7 Energy)).
We would like to thank Vattenfall Wind Power and Senvion SE for making this investigation possible.
Furthermore we would like to thank the R Core Team for developing the open source language R (Team, 2015).





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

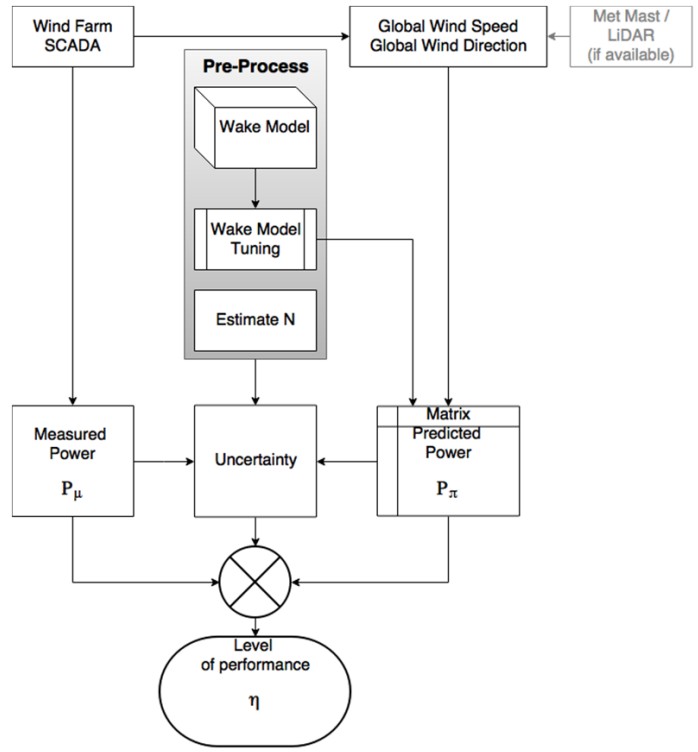

**Figure 1: Flowchart of the Performance Monitoring Model.**





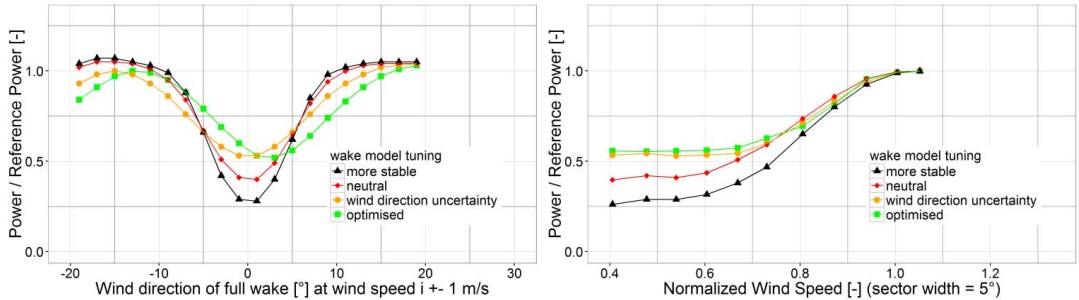

**Figure 2: Impact of different key tuning aspects on the wake model results. An increasing atmospheric stability increases the wake**
5    **deficit (from red rhombus to black triangles). Wind direction uncertainty flattens the wake deficit (orange points), and a wind**
**direction bias shifts the deficit horizontally (green squares).**

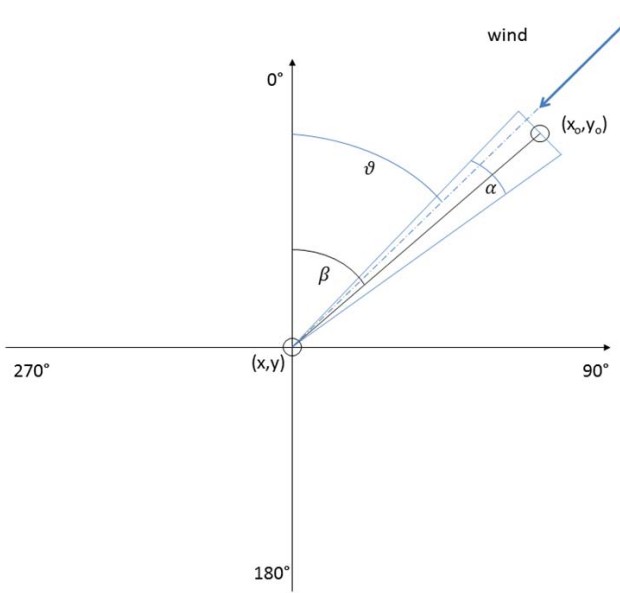

**Figure 3: Determination of the waked sector. The turbine at (x0,y0) produces a wake on the turbine at (x,y) for the displayed wind**
10    **direction ϑ**




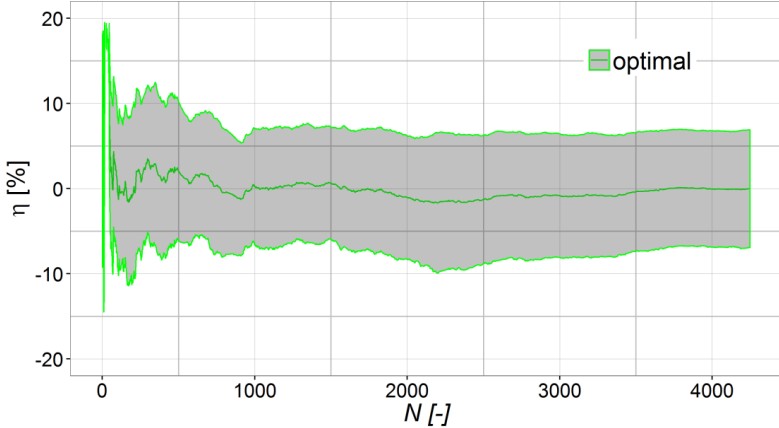

**Figure 4: Underperformance indicator η with uncertainty margin as function of the number of measurement values N. Derived with the calibrated model at a turbine in triple wake.**

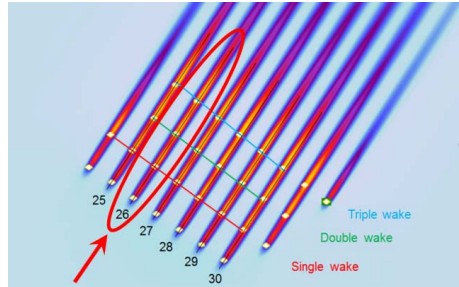

5    **Figure 5: Wind farm Ormonde layout with simulated wakes for a full wake situation from a south-westerly wind direction.**




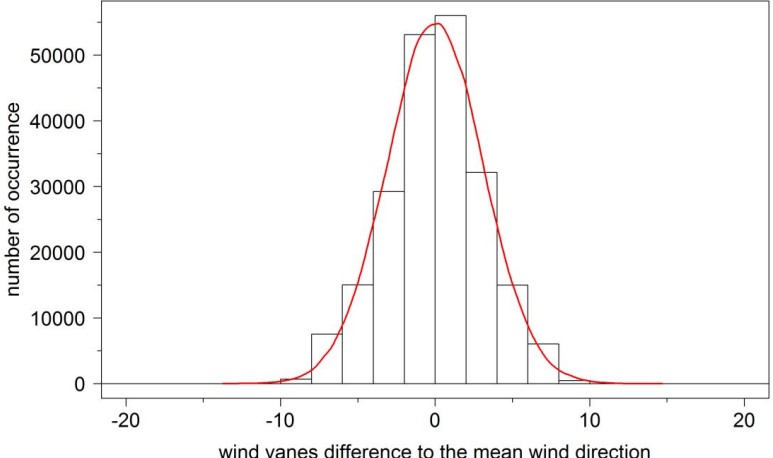

**Figure 6: Histogram of the deviation of 30 individual wind vanes from the average wind direction for the full data set with a sector of 30° centering the full wake condition.**

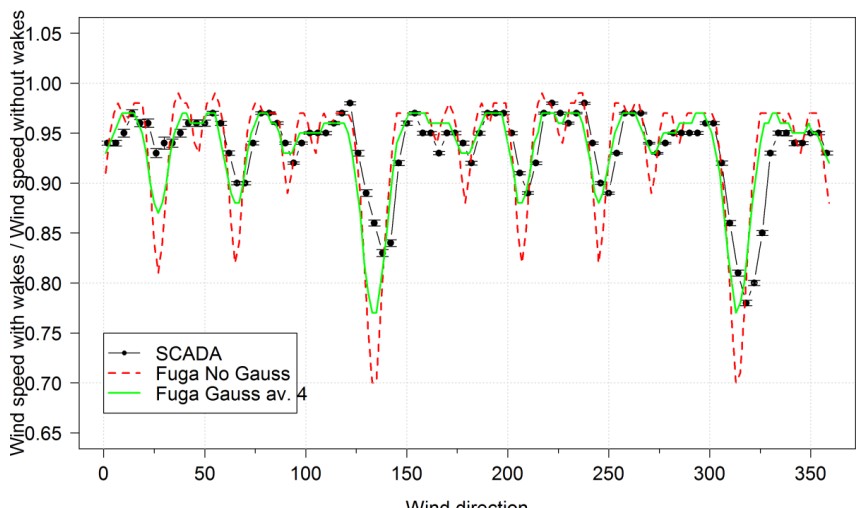

**Figure 7: Wind farm averaged wind speed with wake effects normalised with wind farm averaged wind speed without wake effects plotted versus averaged wind farm wind direction. Black dots show the measurements from SCADA and the green solid line represents the results from Fuga with a Gauss averaging for standard deviation of 4°.**




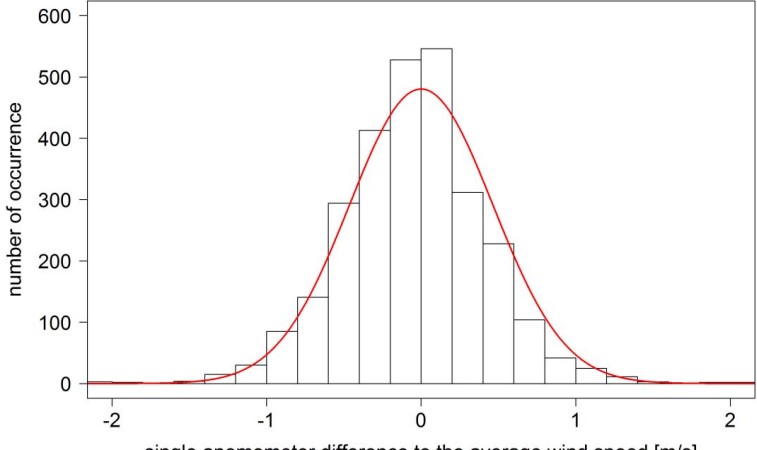

**Figure 8: Histogram of the wind speed difference of a single anemometer to the average wind speed of all anemometers. The displayed Gaussian distribution (red line) has the standard deviation of 0.46 m/s. A sector of 30° centering full wake alignment has been selected.**




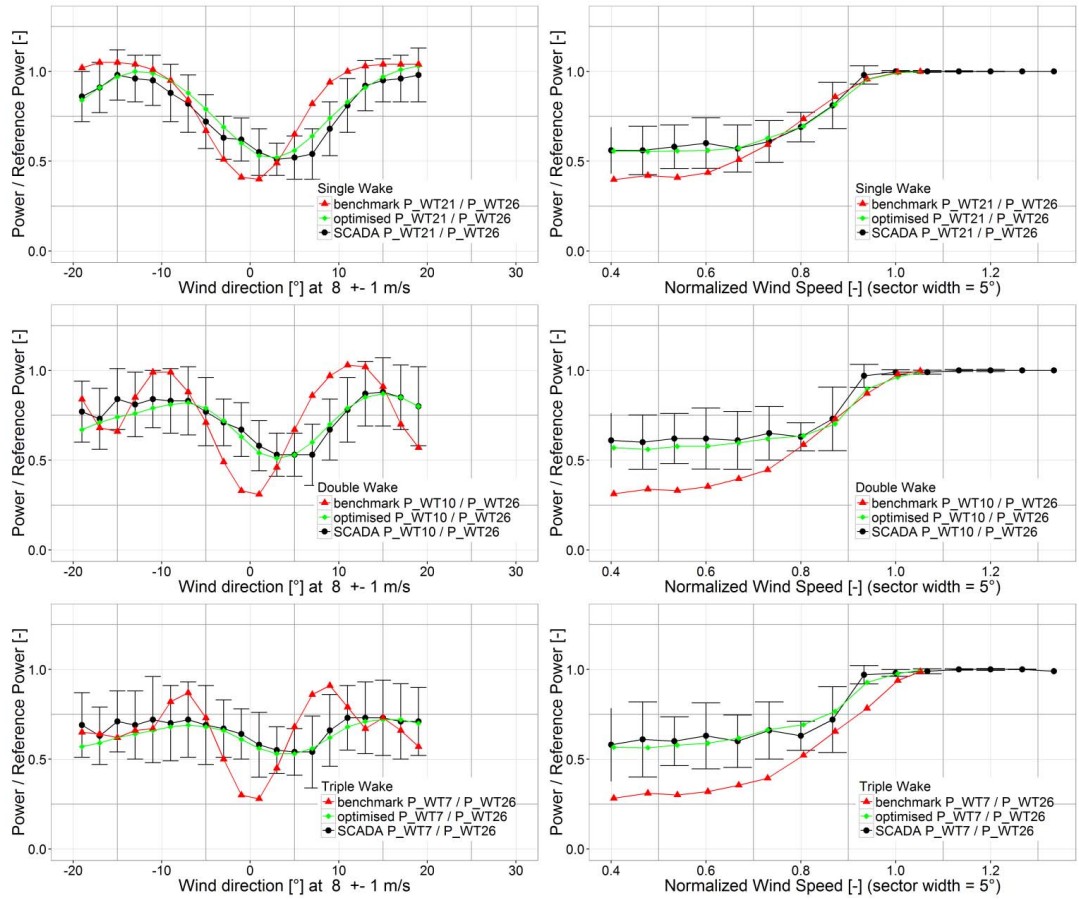

**Figure 9:** (left column) Power normalized by the power of the free flow turbine as function of the wind direction centred at full wake for 8 ± 1 m/s wind speed. (right column) Power normalized by the power of the free flow turbine as function of the wind speed normalized by wind speed at rated power for the waked turbine.





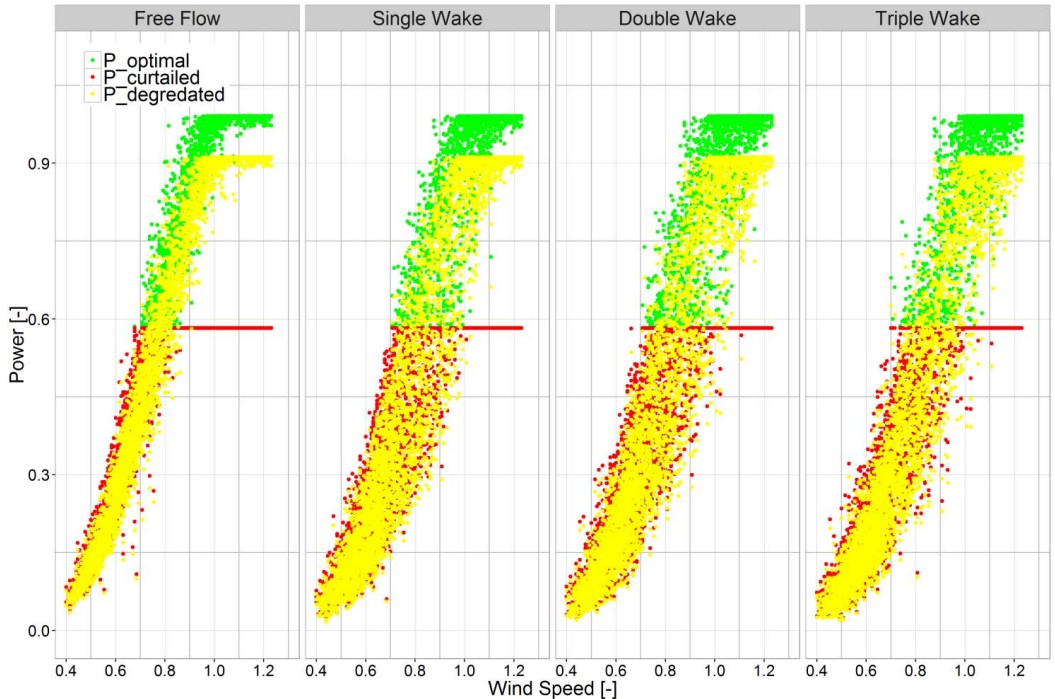

**Figure 10: Scatterplot with normalized power as function of the normalized wind speed for four turbines in one row with two error test cases.**





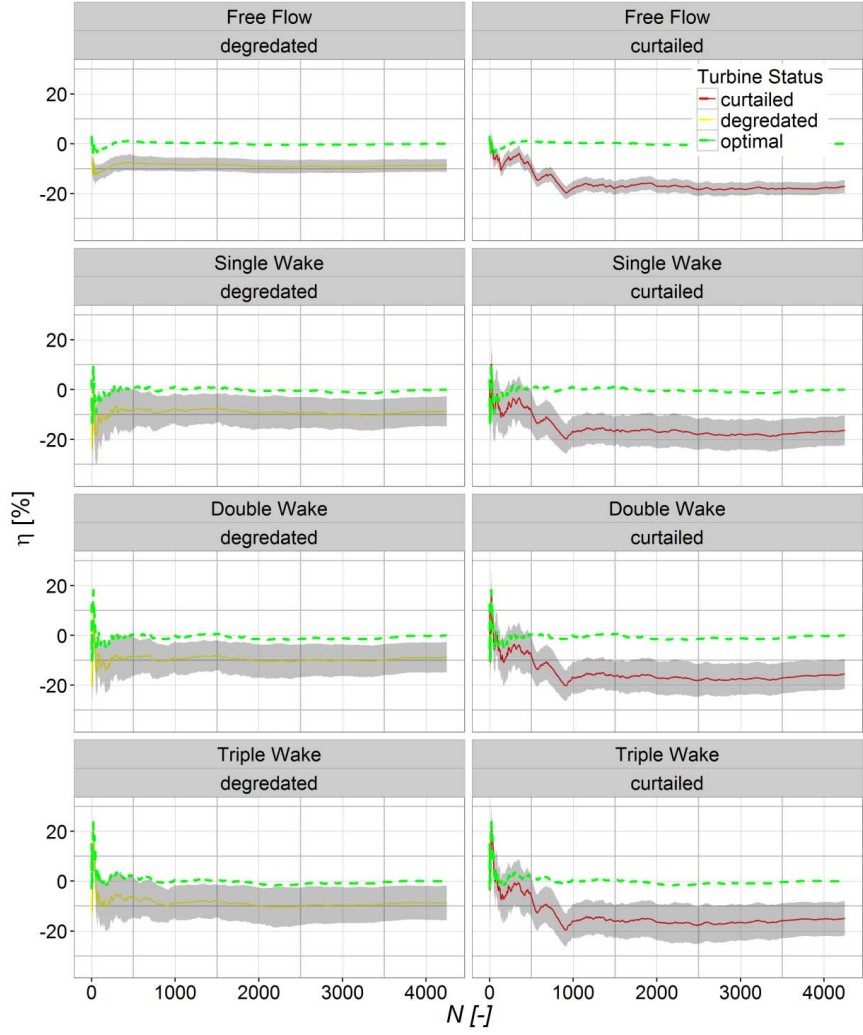

**Figure 11: Underperformance detection for curtailment and degradation at turbines with different levels of wake influence.**





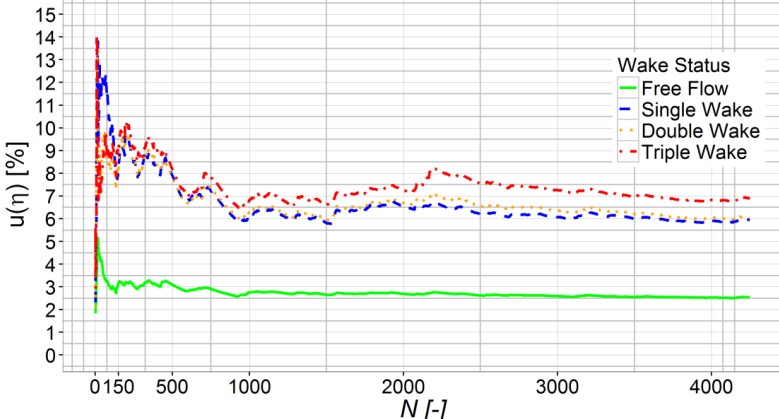

**Figure 12:** Uncertainties for the underperformance indicator $u(\eta)$ as function of $N$ values for free flow, single wake, double wake and triple wake situation.





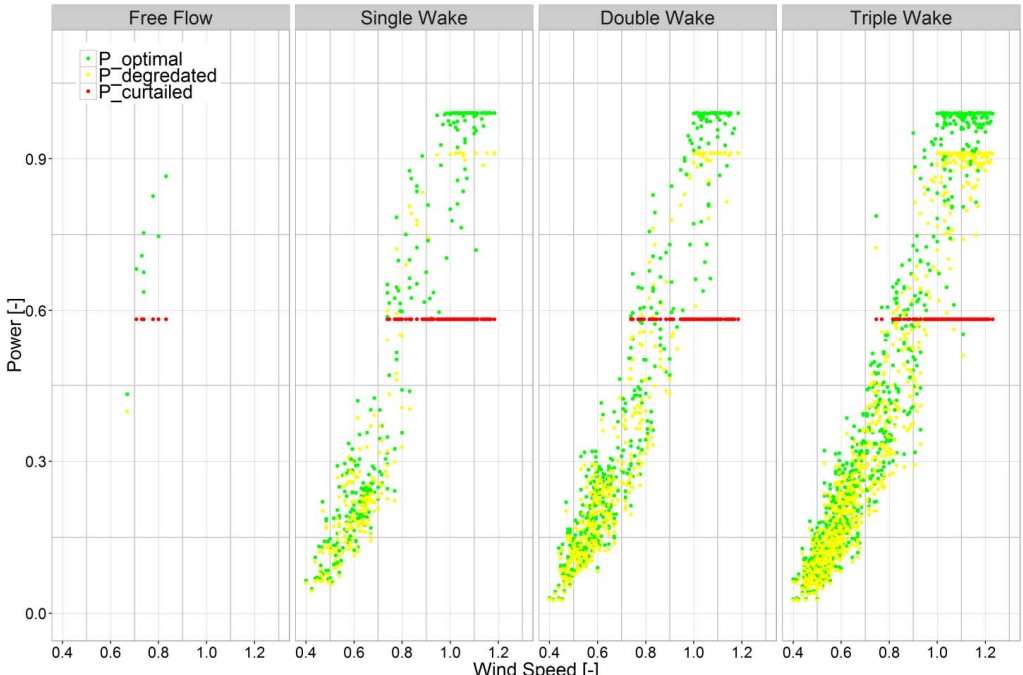

**Figure 13: Scatterplot of each turbines normalized power curve. The quantity N=126 equals to the estimated sample size for the first detection of degradation at a turbine in triple wake situation.**

**Table 1: Type B uncertainties of the predicted power $P_\pi$**

| k | Uncertainty Component | Sensitivity $c_{k,i,j}$ | Uncertainty $u_{k,i,j}$ |
|---|---|---|---|
| 1 | Wind speed estimation | $\left\lvert \dfrac{P_{\pi\,i,j} - P_{\pi\,i-1,j}}{V_{ij} - V_{i-1,j}} \right\rvert$ | One standard deviation of the averaged anemometers |
| 2 | Wind direction estimation | $\left\lvert \dfrac{P_{\pi\,i,j} - P_{\pi\,i-1,j}}{\vartheta_{ij} - \vartheta_{i-1,j}} \right\rvert$ | One standard deviation of the averaged wind direction |

**Table 2: Type B uncertainties of the measured power $P_\mu$**

| k | Uncertainty Component | Sensitivity $c_{k,i,j}$ | Uncertainty $u_{k,i,j}$ |
|---|---|---|---|
| 1 | Current transformer | 1 | 0.0043 P [kW] |
| 2 | Voltage transformer | 1 | 0.003 P [kW] |
| 3 | Power transducer | 1 | 0.003 $P_{rated}$ [kW] |
| 4 | Power data acquisition | 1 | 0.001 $P_{rated}$ [kW] |





**Table 3:** *N* **values to the first detection of underperformance with certainty of one standard deviation. Values in brackets indicate N with wind speeds above the curtailment.**

| Wake situation | N_degradation [-] | N_curtailment [-] |
|---|---|---|
| Free Flow | 1 | 51 (8) |
| Single Wake | 187 | 497 (109) |
| Double Wake | 378 | 498 (103) |
| Triple Wake | 731 | 526 (126) |

5   **Table 4: Uncertainty** *U* **at quantity** *N* **of first detection of underperformance with certainty of one standard deviation.**

| Wake situation | U_degradation [%] | U_curtailment [%] |
|---|---|---|
| Free Flow | 1.8 | 3.6 |
| Single Wake | 8.9 | 7.9 |
| Double Wake | 8.7 | 7.8 |
| Triple Wake | 8.0 | 7.8 |