# Peer review of "Monitoring offshore wind farm power performance with SCADA data and advanced wake model"

_Wind Energy Science, 2016_

## Referee Comment (RC1) · Anonymous Referee #1 · 15 Jun 2016

General comments

I think that the subject in general is interesting as wind farm underperformance is an important issue that we sometimes do not want to discuss much in wind energy. Therefore, I started to read with interest the manuscript but at about the second page I became really bored of the continuous issues/typos/grammatical problems that the text has. It is not that the English is generally bad; it is more about the way the authors write sentences and connect the ideas. It is generally "very weird" the way they write. In the specific comments, I list a number of issues but as I said I became so bored of these things so I just did it for the first pages; in case the authors have the chance to resubmit, the manuscript has to pass many hands including some English technical experts before resubmitting. More important, the manuscript in its actual form reads more like a technical report describing a method rather than a scientific paper. The

authors need to make clear what the contribution to science is (if any) and write the manuscript to establish that the method they suggest is clearly novel (so far I do not see the novelty; the wake model is not new, neither the uncertainty calculation). Also they make things harder to digest by their writing so the text needs some reshuffling to accomplish a good flow.

About the subject: There is a clear shift of the direction of the wake even for the single wake case. The authors provide some arguments but in the single wake case the maximum wake deficit should simply be a 0 deg. The authors use a nacelle-based vane for the wind direction so why not checking if there is a systematic turbine misalignment by looking at the nacelle position signal in the SCADA data? The authors also refer to the study of Vollmer, but in that study, the wake is deflected intentionally by misaligning the turbines. So the most plausible explanation it is simple yaw misalignment unless the authors discard this by showing that the turbines are indeed not misaligned (but they do not do that).

There is a general problem with the way the authors make references in the text and the reference list itself at the last section. You should write refs. in the text as: "A power curve is given for each turbine (Smith, 2001). However, Jonas (2010) described another method. Such method was also shown in some previous studies (Klinsmann, 2006; Pauli, 2010)". In the specific comments I select some specific cases but most of the references are wrongly made. And the references in the reference list should be made consistently: Names, title (non-capital all refs. or all capital), etc. Such type of reference list makes me wonder about the quality of the whole study. The reference list should be made with consistency. Also you have a problem with the equations; they are part of the text and should not disrupt it! The dot symbol does not mean multiplication, it means dot product but you don't have such products. The "same" symbols are sometimes in italics and sometimes in normal text; if they are the symbol of the same thing then they should be written in the same way.

Specific comments:

1. Page 1 line 16 "...technical solutions." This type of statements are very general and not precise and specific. What do you mean by this? Turbines, models, methods?

2. Page 1 line 18 "...definition (...) defines..." that the system is ready to operate" this is redundant. Why not "...definition (...) is that related to a system ready to operate"

3. Page 1 line 20 "... quality and quantity." Of what? In the next paragraph you kind of explain it but you cannot simply say this here and expect that the reader finds the answer later. If this is the case then that sentence can be removed.

4. Page 1 line 21 Replace "...much SCADA" by "lots of" or "a good amount of"

5. Page 1 lines 27-28 "Work on... of 2016". You don't need this reference and does not help the paper so remove it

6. Page 1 lines 28- Page 2 line 1 "For most turbines. ... wake effects" You make it sound as it was only a problem for offshore turbines and it is not so replace by e.g. "For most turbines in a typical wind farm, verification of the performance by comparison with the power curve is not suitable..." then "... maintenance of a met mast is very expensive particularly offshore."

7. Page 2 line 3 Replace "accounts" by "account"

8. Page 2 line 6 "Incorrect parameter settings" you mean "turbine parameter"?

9. Page 2 line 9 "turbine has a limited power output which has been externally applied" You want to refer to the limit but with the use of the "which" you mean the limited power output but surely this is not what is externally applied because one cannot apply a limited power output... that is a consequence of limiting something else.

10. Page 2 lines 11-16 this is a very weird paragraph. "Upwind turbines influencing the free flow for downwind turbines". This is a weird sentence because the flow is not free for the downwind turbines. Why not just removing the "free" word. Then it is also weird because you have "..., wake effects," so with this construction you say the wake

effects are turbines! Then you have these references to Albers (all wrongly made; see my major comment). Then you say "Albers has also looked. . .flow models": this is a personal communication or is in one of his studies? Then comes "But at that time. . ." what time? Which year or which study in particular? There is also a "to be further development" that should be "to be further developed"

11. Page 2 line 18 ". . .which proposed" so the standard stopped at some point proposing this?

12. Page 2 line 19-20 ". . .,could not be established," So it is not a standard, it is a working group trying to come up with a standard. Also delete the part ". . .and the support. . . crumbled." It is not scientific knowledge

13. From line 22 in page 2 onwards you talk about "matrices" but what you mean is "look-up-tables (LUT)" Use that term. There is an unnecessary comma in line 22 after "method". Also in that line you talk about detection of "curtailment". Perhaps your method is able to detect curtailment but curtailment is generally artificially imposed or used and so it is/should be recorded in the turbine status of the SCADA.

14. Page 2 lines 23-27 "Mittelmeier et al. (2013) presented. . .. environmental conditions." I am not sure this is a new method. In many other studies, authors used wake-model-based LUTs to estimate the efficiency of the wind farm. So the authors need to explicitly say what exactly is new.

15. Page 2 lines 27-28 "Especially. . . available." Already mentioned so remove it

16. Page 2 lines 31-33 Replace "this" by "the", add "of" after "method" and use "Mittelmeier et al. (2013)" instead of "(Mittelmeier et al., 2013). Replace "condition" by "conditions"

17. Page 3 line 1-2 "Hence the presented. . . LiDAR". Based on what you have already mentioned one can inferred what is written here so it is not necessary

18. Page 3 line 4 it should be change to ". . ..of the method by Mittelmeier et al. (2013)

is recalled".

19. Page 3 line 11 I know what you mean by "deviation" but you need to be exact so change to "A deviation between P\pi and P\mu"

20. Page 3 lines 14-15 "The accuracy and calculation …. real wind farm flow" This is not true. A simple wake model can be as accurate as a complex one.

21. Page 3 line 16 First "improve the underperformance detection capabilities" This is not always true. And replace "," by ";" before "on the other hand"

22. Page 3 line 23 Remove the first "Additional"

23. Page 3 line 25 "for this purpose" I know what you mean but you have not mention any purpose and you want to refer to the monitoring method, I guess. So be precise

24. Page 3 line 28 Replace "… averages of 10 minutes" by "… averages over 10-min periods"

25. Page 3 line 29 Replace "… averaging N quantities of 10 minutes time samples" by "averaging a number N of 10-min samples"

26. Page 3 line 30 Remove "the averaging"

27. Page 4 line 1 You are talking about "correlation" but this is not a correlation of power it is a only a normalization. So as this is wrong the part of "This leads" does not make sense

28. Page 4 line 4 "This is described in Eq. (1) and (2)" Well this is not described in the equations; the equations are simply the definition you are using for the normalized powers

29. Page 4 line 12 Replace "can be described with Eq. (3)" by "is defined as"

30. Page 4 lines 16-17 "where \eta_ob,ref… (ob) turbine" You already defined everything so there is no need for this

31. Page 4 lines 26-29 So why do you have to use all the wind vanes (this is what is read from the text)? They could all have a different misalignment and so you will need to analyze each of them (in terms of wake deficits) if you want to use all of them.

32. Page 4 line 29 "complex area" what do you mean by complex area?

33. Page 5 line 1 "of the scale" what do you mean by scale?

34. Page 5 line 2 "+-1.5IQR" be explicit. If the outliers removed are those outside the range +-1.5IQR then say so

35. In Eq. 4 you have constants without units. If alpha is in degrees all these constants have the units of degrees and you need to state that

36. Eq. 6 is not needed

37. Page 6 line 1 "the north inconsistency need different conditions" Yes obviously

38. Page 7 lines 1-2 "In Mittelmeier et al. (2015)... wind speeds". Well that depends on the stability conditions. This will be true if compare unstable conditions with low wind speeds and high sigmas with neutral conditions with lower sigmas and higher wind speeds. But stable conditions will be in the low wind speed range with lower sigmas compared to neutral

39. Page 7 line 3 "no impact" you mean "little impact"

40. Page 7 lines 7-8. This seems to be quite important and you do not provide any details about the study of Vollmer et al.

41. Page 7 line 31 "usually about 4 to 6%" you need to give a reference here; otherwise show an example

42. Page 8 line 19 "is around 7%" based on what?

43. Page 9 line 3 Figure 5: please show a proper layout with north orientation and scales

44. Page 9 line 4 "6.3D" this is between turbines in the same row (does not look like that), between rows? And important what are rows for you: the rows of turbines in a particular direction or all "rows" of turbines?

45. Page 9 line 7 Two years of "10-min" SCADA data?

46. Page 9 line 10 "The quality of the derived wind direction" It is not the quality what you show there

47. Page 9 line 16 "from Section..." Explicitly state which equations

48. Page 9 line 18 "binned into 2 deg" does not look like that but more like 4 deg.

49. Page 9 line 23 These correlations are very high but the outliers seem to be also quite large so I am very skeptical about these computations

50. Page 9 The wake model seems to have a systematic bias when compared to the measurements (clearly seen at the highest deficits). Why? Is the average wind direction perhaps wrong? Perhaps you should average the wind direction of the turbines in the rows where the flow is not disturbed if you want to compare it with the wake model

51. Page 10 line 1 The wake model should be presented before the results in 3.1.2!

52. Page 10 line 11 How much data you use for the calibration period?

53. Page 10 starboard and port terms are terms conventionally used in wake studies?

54. Page 10 line 17 what do you mean by "global"

55. Page 10 lines 16-22 If this phenomenon occurs in 1 single wake the a very plausible reason is simply yaw misalignment

56. Page 10 line 32 "this data has been filtered for a wind direction sector of 5 deg" Change to "These data have been.."

57. Page 11 line 2 You mean for prediction of what? AEP? Wind speed? A particular case?

58. Page 11 line 5 Replace "has been" by "will be"

59. Page 11 line 6 what do you mean by "real data"? so before the data was not real?

60. Page 11 line 9 degradation of 8% in terms of what?

61. Page 11 line 13 Replace "in displayed" by "is shown"

62. Page 11 line 18 Why "Therefore"?

63. Page 12 line 10 "horizontal graph" you mean horizontal line?

64. Symbols in Fig. 1 are not the same symbols. . . they are not in italics

65. Figure 3: The wind direction should not be perpendicular to the rotor? \beta (which is not the wind direction) is the angle perpendicular to the rotor

66. Figure 5: scales, north, coordinates!

67. Figure 10: there should be some green points below 0.6, so perhaps it is better to degrade based on the best Cp curve

---

## Referee Comment (RC2) · Anonymous Referee #2 · 16 Jun 2016

Paper: wes-2016-16 Title: Monitoring offshore wind farm power performance with SCADA data and advanced wake model Authors: Niko Mittelmeier et al.

It is an interesting paper, introducing a new validation method for identifying wind farm underproduction. Such methods are highly needed with the large amount of wind turbines are installed in wind farms. The precondition for my review is that the method should also be applicable for implementation and not only be an theoretical exercise.

The method, which seems to be a spin-off from an EERA project named ClusterDesign, refers to an ideal determination of the inflow conditions. The proposed method uses wake models estimates as reference, which seems to make a robust estimate of the underproduction. The accumulated uncertainty for the inflow conditions has been estimated to 7% and this number seems realistic when using recent calibrated instruments (cup and vane). This number is not realistic when using derived inflow conditions based on nacelle anemometry, electric power and wind turbine yaw position for periods longer than 1 year according to my experience. Problem: The determination of the wind farm inflow (environmental) conditions (wind speed and wind direction) seems not to be aligned with the state-of-art wind farm signals.

In section 2.1.1 the wind direction is derived, but without any reference to how this is done. The wind direction measured on the nacelle is only used for yaw control, where the strategy is the keep the rotor aligned with the wind direction to minimize the yaw-misalignment. This signal can also identify a "forced" yaw misalignment used to determine the "wake drift"? The optimal readings from this instrument is 0, and will not reveal anything about the actual flow direction, which only can be identified from the wind turbine yaw position. The wind turbine yaw position not used by the controller, only when wind farm has sector management (proposed but never seen). The yaw position is usually not calibrated or has a wrong offset, which need to be identified.

Section 2.1.2 states to use nacelle anemometry to determine the wind speed; correct this is the only accessible wind speed measured on a wind turbine. This is measured with either a cup anemometer or sonic, located on the nacelle (behind the rotor). The signal is recorded through the controller and stored in SCADA system, but lacks documentation and uncertainty estimation. A correlation check between a number of identical wind turbines reveals a large scatter in the binned power curves. The scatter increases when the turbine operates in a wake compared to free inflow. Conclusion: the nacelle wind speed signal is biased. Furthermore the nacelle anemometer changes over years e.g. due to degradation. Even a NTF based wind speed (IEC 61400-12-2) is only applicable for free, undisturbed inflow.

Conclusion on inflow conditions: the stated uncertainty, for wind speed and wind direction does not meet the requirements given in IEC 61400-12-1 and this need to be addressed both in the method and in the example. Comments to the figures: all figures should include proper captions readable out of context. The caption of the figures are

not sufficient e.g. while Figure 2b is not a addressed in the caption.

The description of the method seems to be adequate, but the "wake drift" in section 2.2 is not well defined, I assume this refers to periods with active wake control, which I do not expect has been implemented yet?

---

## Referee Comment (RC3) · Anonymous Referee #3 · 23 Jun 2016

Dear Authors,

thank you for a interesting and novel idea for the detection of underperforming wind turbines. While I agree with the overall tenet of the paper, there are a few issues I would like to have clarified. Particularly, the SCADA system delivers data at a much higher rate than the 10-min averages. What would happen if you'd make use of the 1-sec resolution available from the data? 130 values would suddenly be 2 minutes instead of 21 hours, if every assumption stays unchanged - which it probably doesn't. How do you deal with intra-10-min variability? Do you require a relatively stable weather situation or at least wind direction to be able to do it?

More detailed comments:

Page 2 Line 1: Offshore met masts are very expensive. Are people really putting up

met masts offshore to verify the turbine performance? Or onshore?

P3L17: I think this is debatable. Since you need quite a number of values / quite some time to detect the deviations, the method is not really real time anyway, so it could also be analysed retroactively every now and then. It also could be run on a larger high-performance computer based on downloaded SCADA data. Besides, the connection between higher computational cost and accuracy of the wake models is sketchy at best, see e.g. the results presented in WindBench.

2.1.1: I wonder why you do not use the same correction method on all the wind farms wind vanes? It just requires SCADA data and some computer power, so it would not be too difficult. Also, when you're calculating a mean wind direction for the whole farm, aren't you relying on a smaller size farm far away from the coast? For example, if you have a location like Anholt, then you have wind and direction gradients due to the proximity of the land across the wind farm. How would that influence your method and its accuracy?

P4L1: Is that one reference turbine for the whole farm, or one particular one for each of the other turbines? If it is one for the farm, how is it defined?

P7L23: So the Type A uncertainties do not multiply in a multipe wake situation?

P10L15-22: This is quite interesting. Has this behaviour been observed anywhere else? Another explanation could be that the overall wind flow is skewed at the Or-monde location, which judging by the map is not impossible, seeing that the wind farm is wedged between land and a larger offshore wind farm. Or do I understand this wrong, and it is an effect from Fuga which is described here? Did you switch on the meandering mechanism in Fuga?

Figure 5: A map of the location would be good here (see above).

Figure 7: There seems to be a shift in wind direction between the SCADA system and the calculations - any idea where that is coming from?

Table 2: Is the source of that the IEC Annex D uncertainty estimation, or own values?

Textual issues:

P1L9: The presented method or the present method?

P1L18: "The common [...] definition defines that the system is ready to operate" might not be exactly what is in the availability standard. Please rephrase.

P1L23: What's the difference between IEC 61400 and IEC TS 61400?

P2L25: To lower uncertainties, ... ??

P6L15: You might want to explain what you mean by "wake drift".

P9L9/L15: "of demonstration wind farm" could be deleted without detriment.

P9L13: This value is used_for the uncertainty...

P10L5: I would drop the brackets around the version number of Fuga.

P10L14: I don't think I've seen zeta_naught introduced before?

P10L15: While wind and nautical terms can easily be construed to have a connection, not everyone is familiar with "starboard".

Figure 1 center: Should that really be called Uncertainty, or rather something like Deviation?
* * *

---

## Author Comment (AC1) · 26 Jul 2016

Paper: wes-2016-16
Title: Monitoring offshore wind farm power performance with SCADA data and advanced wake model
Authors: Niko Mittelmeier et al.

Answers to comments of anonymous Referee #1 by
Niko Mittelmeier et al. July 26, 2016

Dear Referee,

Thank you very much for your honest feedback and the very detailed and helpful comments. Your advices and suggestions will certainly help to make the paper clearer and better. Below we have addressed each comment. Your overall message, "be more precise" is understood and we hope to meet your expectations. Our response is marked as follow: ***/ response /***

**General comments**
I think that the subject in general is interesting as wind farm underperformance is an important issue that we sometimes do not want to discuss much in wind energy. Therefore, I started to read with interest the manuscript but at about the second page I became really bored of the continuous issues/typos/grammatical problems that the text has. It is not that the English is generally bad; it is more about the way the authors write sentences and connect the ideas. It is generally "very weird" the way they write. In the specific comments, I list a number of issues but as I said I became so bored of these things so I just did it for the first pages; in case the authors have the chance to resubmit, the manuscript has to pass many hands including some English technical experts before resubmitting.

***/ Introduction will be rewritten. Focus: "be more precise" (see comments below) /***

More important, the manuscript in its actual form reads more like a technical report describing a method rather than a scientific paper. The authors need to make clear what the contribution to science is (if any) and write the manuscript to establish that the method they suggest is clearly novel (so far I do not see the novelty; the wake model is not new, neither the uncertainty calculation). Also they make things harder to digest by their writing so the text needs some reshuffling to accomplish a good flow.

***/ For our wind farm monitoring model we have chosen the power matrix approached (look-up tables (LUTs)) which has been used in several studies before. (TC88 WG6, 2005 ), (Mellinghoff , 2007), (Carvalho and Guedes 2009) , (Westerhellweg et al., 2012) and (Mittelmeier et al. 2013).

We see the advantage of LUTs in the fact, that any wake model can be chosen to provide input for our model. And with further improved wake models, the monitoring method will improve.

The novelty is a new turbine referencing approach. Not the absolute values between model and measurement are compared, but the relation between an observed turbine and all other turbines in the farm. The uncertainty of the resulting performance ratio is much lower than the uncertainties of absolute production or AEPs. Furthermore all the above mentioned publications have proposed to use met mast data and we have demonstrated our method only with measurements which are available on state of the art wind turbines (SCADA data).

Usually measured data from nacelle mounted devices is prone to errors due to disturbed flow behind the rotor. When looking at absolute power values this would lead to high uncertainties. The IEC 61400-12-2 (2013) standard provides an example in Annex J showing uncertainties of approximately 20% on AEP for one turbine. A reduction in AEP uncertainty could be achieved by multiple measurements.

By using reference turbines, uncertainties from air density corrections can be neglected. Furthermore, the uncertainty of the wind speed has a much lower sensitivity factor. Wind direction measurements have a clear contribution to the combined uncertainties in our model, but we could show, that in our example, 7% uncertainty on the performance ratio are an improvement compared to existing methods.

The contribution to science is an explicit investigation on how underperformance can be detected in single wake, double wake and triple wake situations and we provide validation and suggestions how to improve results of the selected wake model in pre and post process. We have used Fuga because it is accessible, fast and easy to handle. But for experienced users of other models, the choice might be different and that's ok for our model.

**References:**

Carvalho, H. and Guedes, R.: Wind Farm Power Performance Test , in the scope of the 2 . Wind Farm Performance Matrix, , 1–9, 2009.

IEC 61400-12-2: Power performance of electricity-producing wind turbines based on nacelle anemometry., 2013.

Mellinghoff, H.: Wind Farm Power Performance Verification, DEWI., 2007.

Mittelmeier, N., Amelsberg, S., Blodau, T., Brand, A., Drueke, S., Kühn, M., Neumann, K. and Steinfeld, G.: Wind farm performance monitoring with advanced wake models, in Proceedings of the EuropeanWind Energy Association Offshore Conference, Frankfurt., 2013.

TC88 WG6, I.: Wind farm power performance testing working group draft, IEC., 2005.

Westerhellweg, A., Canadillas, B., Kinder, F., Neumann, T. and Windenergie-institut, D. G. D.: Farm Efficiency and Power Matrix based on RANS ( CFD ) Simulations for the offshore Wind Farm Alpha Ventus and Comparison with Measurements, 2012.

/***

About the subject: There is a clear shift of the direction of the wake even for the single wake case. The authors provide some arguments but in the single wake case the maximum wake deficit should simply be a 0 deg.

***/ We provide you with some recent publications which support the theory, that even in the wake of a single turbine with now yaw error a shift of the wake is observable. These studies have the general aim to investigate active wake control but they also provide examples for 0°yaw error. Fleming (2013) shows in his baseline simulation (no yaw error) a small wake shift to the right when looking downwind. In the LES study of Vollmer et al. (2016) it can be observed, that the wake deflection increases from neutral (Vollmer et al. 2016, Fig. 5) to stable conditions(Vollmer et al. 2016, Fig. 9). These Figures provide simulated results also for a turbine with 0° yaw angle. In both cases the maximum wake deficit is found to be on the right side of the centre line (looking downstream).
Gebraad (2014, p86) gives an explanation for the observations from the simulations by Fleming (2013). The flow reacting on the rotation of the rotor causes the wake to rotate counter clockwise (looking downstream). Higher wind speeds from the upper layer are transported downwards (on the left side) and lower wind speeds from the lower layer are pushed upward on the right side of the wake. As a result the velocity deficit at the right part of the wake increases, so the wake deflects to the right.
Marathe et al. (2015) could show in their field measurement campaign with dual-doppler radar, that in the near wake region, the wake is drifting to the right, as expected by the theory. But in the far wake they registered a contradicting movement. The authors state the hypothesis that this phenomenon may be caused by atmospheric streaks. An offshore field experiment by Beck et al. (2015) provides further evidence that wakes are moving out of the centre line.

**References:**

Beck, H., Trujillo, J. J., Wolken-möhlmann, G., Gottschall, J., Schmidt, J., Peña, A., Gomes, V., Lange, B., Hasager, C. and Kühn, M.: Comparison of simulations of the far wake of alpha ventus against ship-based LiDAR measurements, in RAVE Conference., 2015.

Fleming, P. A., Gebraad, P. M. O., Lee, S., van Wingerden, J. W., Johnson, K., Churchfield, M., Michalakes, J., Spalart, P. and Moriarty, P.: Evaluating techniques for redirecting turbine wakes using SOWFA, in ICOWES2013 Conference., 2013.

Gebraad, P. M. O.: Data-Driven Wind Plant Control, 2014.

Marathe, N., Swift, A., Hirth, B., Walker, R. and Schroeder, J.: Characterizing power performance and wake of a wind turbine under yaw and blade pitch, , doi:10.1002/we, 2015.

Vollmer, L., Steinfeld, G., Heinemann, D. and Kühn, M.: Estimating the wake deflection downstream of a wind turbine in different atmospheric stabilities: An LES study, Wind Energy Sci. Discuss., (March), 1–23, doi:10.5194/wes-2016-4, 2016.

/***

The authors use a nacelle-based vane
for the wind direction so why not checking if there is a systematic turbine misalignment
by looking at the nacelle position signal in the SCADA data?

***/ For our monitoring model we are using the absolute wind direction signal from each turbine which is defined as

$$\vartheta = nacelle\ position + wind\ vane\ position$$

The nacelle position is the angle between the rotor axis and a marking for true north. This marking is calibrated as part of the commissioning. But often this signal is not maintained well during operation, because it has no effect on turbine performance. This causes the necessity to apply a bias correction to this signal before using it for reanalysis purposes. The wind vane position indicates the angle of the flow to the rotor axis. It directly provides a value for the yaw error. The turbine controller  uses this signal to control the yaw activity. For an infinite averaging the mean value of the wind vane position is 0°. We have used 12127 10-min values for the wake model calibration. A histogram of the vane position signal for the whole data is provided in  Figure x below.

[Figure]

Figure x: Histogram of the wind vane position signal from turbine 26 showing all data that has been used for wake model calibration. Resulting in a mean value of -0.2 and a median of -0.3.

A systematic yaw error resulting from the sensor alignment can be estimated to ±2° for a single turbine (IEC 61400-12-2, 2013). A nacelle transfer function for wind direction is used to take the effects of the rotor into account. We do not assume this correction being perfect, so we end up with uncertainties of approximately 3°.

We will include these clarifications in Section 2.1.1. And add further explanations to the data handling, filtering and corrections that are necessary to obtain the right quality signal. (See also Comment 64)

**Reference:**

IEC 61400-12-2: Power performance of electricity-producing wind turbines based on nacelle anemometry., 2013.

/***

The authors also refer to
the study of Vollmer, but in that study, the wake is deflected intentionally by misaligning the turbines. So the most plausible explanation it is simple yaw misalignment unless the authors discard this by showing that the turbines are indeed not misaligned (but they do not do that).

***/ In the study of Vollmer, simulations are showing wake behaviour for 30°, 0° and -30° yaw angle under different atmospheric conditions. Even at 0° yaw angle the maximum wake deficit is not at the centre line (perpendicular to the rotor). This effect increases with increasing atmospheric stability.

Unfortunately we don't have information from the site which would allow a stability classification based on the Monin - Obukhov length. But we obtained the best wake deficit fit between the SCADA data and the Fuga calculations with a $\zeta_0 = 2.72\,e - 7$ which is supposed to be used for more stable cases.

We have changed the reference to the following publication:
Vollmer L, Steinfeld G., Heinemann D., Kühn M.: Estimating the wake deflection downstream of a wind turbine in different atmospheric stabilities: An LES study, doi:10.5194/wes-2016-4, 2016

/***

There is a general problem with the way the authors make references in the text and the reference list itself at the last section. You should write refs. in the text as: "A power curve is given for each turbine (Smith, 2001). However, Jonas (2010) described another method. Such method was also shown in some previous studies (Klinsmann, 2006; Pauli, 2010)". In the specific comments I select some specific cases but most of the references are wrongly made. And the references in the reference list should be made consistently: Names, title (non-capital all refs. or all capital), etc. Such type of reference list makes me wonder about the quality of the whole study. The reference list should be made with consistency.

Also you have a problem with the equations;
they are part of the text and should not disrupt it! The dot symbol does not mean multiplication, it means dot product but you don't have such products.

The "same" symbols are sometimes in italics and sometimes in normal text; if they are the symbol of the same thing then they should be written in the same way.

***/ Thanks for pointing this out. It will be improved accordingly. /***

**Specific comments:**

1. Page 1 line 16 ": : :technical solutions." This type of statements are very general and not precise and specific. What do you mean by this? Turbines, models, methods?

***/ The intention was to start the paper with a very generic statement cause not only good turbines will make an investment successful. Installation, O&M, grid components, models, monitoring methods and guarantees are also important. But we agree, being more precise here will help the reader.

We will change the wording from "technical solution" to "wind turbine" /***

2. Page 1 line 18 ": : :definition (: : :) defines: : :" that the system is ready to operate" this is redundant. Why not ": : :definition (: : :) is that related to a system ready to operate"

***/ New wording:
"In wind industry, the common standard IEC TS 61400-26-1 (2011) defines different categories of turbine conditions and describes the calculation of availability". /***

3. Page 1 line 20 ": : : quality and quantity." Of what? In the next paragraph you kind of explain it but you cannot simply say this here and expect that the reader finds the answer later. If this is the case then that sentence can be removed.

***/ You are right. The sentence is moved to the next paragraph. /***

4. Page 1 line 21 Replace ": : :much SCADA" by "lots of" or "a good amount of"

***/ "much" replaced by "lots of" /***

5. Page 1 lines 27-28 "Work on: : : of 2016". You don't need this reference and does not help the paper so remove it

***/ Sentence is removed. /***

6. Page 1 lines 28- Page 2 line 1 "For most turbines: : :. wake effects" You make it sound as it was only a problem for offshore turbines and it is not so replace by e.g. "For most turbines in a typical wind farm, verification of the performance by comparison with the power curve is not suitable: : :" then ": : : maintenance of a met mast is very expensive particularly offshore."

***/ New wording:
"For most turbines in a typical wind farm, verification of the performance by comparison with the power curve is not suitable due to wake effects. And the installation and maintenance of a met mast is very expensive particularly offshore." /***

7. Page 2 line 3 Replace "accounts" by "account"

***/ replaced /***

8. Page 2 line 6 "Incorrect parameter settings" you mean "turbine parameter"?

***/ New wording: "Incorrect turbine parameter settings…" /***

9. Page 2 line 9 "turbine has a limited power output which has been externally applied" You want to refer to the limit but with the use of the "which" you mean the limited power output but surely this is not what is externally applied because one cannot apply a limited power output: : : that is a consequence of limiting something else.

***/ New wording:
"A curtailed turbine has a limited power output below its expected power. Possible reasons for curtailments are load reduction or grid requirements. For these incidents, turbine parameters are changed on purpose and therefore documented in the turbines SCADA logs." /***

10. Page 2 lines 11-16 this is a very weird paragraph. "Upwind turbines influencing the free flow for downwind turbines". This is a weird sentence because the flow is not free for the downwind turbines. Why not just removing the "free" word. Then it is also weird because you have ": : :, wake effects," so with this construction you say the wake effects are turbines! Then you have these references to Albers (all wrongly made; see

my major comment). Then you say "Albers has also looked: : :flow models": this is a personal communication or is in one of his studies? Then comes "But at that time: : :" what time? Which year or which study in particular? There is also a "to be further development" that should be "to be further developed"

***/ The whole paragraph has been revised to be more precise:
"Albers (2004) has published two methodologies for wind turbine performance evaluation. His integral model uses available wind conditions from the energy production of neighbouring WTs, met masts or a combination of both and transfers the information via flow modelling and wake modelling to the investigated wind farm. The measured yield is corrected for turbine availability and then compared against the modelled yield in absolute values. Due to high uncertainties this method is only proposed as a first general check. To reveal smaller deviations he proposes a relative wind turbine performance evaluation model. For this method, active power of direct neighbours are plotted against each other and by comparing two periods, changes can be evaluated. This method explicitly excludes the sectors where wakes are effecting one of the two turbines."

**References:**
Albers, A.: Relative and Integral Wind Turbine Power Performance Evaluation, in EWEC, London., 2004.

/***

11. Page 2 line 18 ": : :which proposed" so the standard stopped at some point proposing this?

***/ The whole paragraph has been revised:
"An international working group (IEC TC88 WG6, 2005) was trying to come up with a standard for wind farm power performance testing. The proposed method uses one or more met masts to establish a measured wind farm power curve matrix. This two dimensional measured power matrix (Wind direction, Wind speed) is compared against a modelled power matrix taking wake effects into account (Mellinghoff, 2007)(Carvalho and Guedes, 2009). The standard could not be established."

**References:**
Carvalho, H. and Guedes, R.: Wind Farm Power Performance Test , in the scope of the IEC 61400-12-3, in Wind Power Expo, Chicago., 2009.

IEC TC88 WG6: Wind farm power performance testing working group draft, IEC., 2005.

Mellinghoff, H.: Wind Farm Power Performance Verification, DEWI., 2007.

/***

12. Page 2 line 19-20 ": : :,could not be established," So it is not a standard, it is a working group trying to come up with a standard. Also delete the part ": : :and the support: : : crumbled." It is not scientific knowledge

***/ Paragraph has been revised. Please see comment 11/***

13. From line 22 in page 2 onwards you talk about "matrices" but what you mean is "look-up-tables (LUT)" Use that term. There is an unnecessary comma in line 22 after "method". Also in that line you talk about detection of "curtailment". Perhaps your method is able to detect curtailment but curtailment is generally artificially imposed or used and so it is/should be recorded in the turbine status of the SCADA.

***/ TC88 WG6 (2005 ), Mellinghoff (2007), Carvalho and Guedes (2009) and Westerhellweg et al. (2012) have used the terminology of "Power Matrix" in their publications. We will give the proper explanation of power matrices being "Look-Up Tables, LUTs" in the introduction and use from there on the terminology of LUTs.

You are right. "Curtailments" are usually recorded in the turbine status of the SCADA. There is no explicit need for the method to detect these behaviour. But we think that redundancy is valuable in performance monitoring and we thing it is helpful to have a second example of underperformance that can be detected by this method.

"or curtailments" has been removed

The comma in line 22 has been deleted. Thank you.

**References:**

Carvalho, H. and Guedes, R.: Wind Farm Power Performance Test , in the scope of the IEC 61400-12-3, in Wind Power Expo, Chicago., 2009.

IEC TC88 WG6: Wind farm power performance testing working group draft, IEC., 2005.

Mellinghoff, H.: Wind Farm Power Performance Verification, DEWI., 2007.

Westerhellweg, A., Canadillas, B., Kinder, F., Neumann, T. and Windenergie-institut, D. G. D.: Farm Efficiency and Power Matrix based on RANS ( CFD ) Simulations for the offshore Wind Farm Alpha Ventus and Comparison with Measurements, 2012.

/***

14. Page 2 lines 23-27 "Mittelmeier et al. (2013) presented: : :. environmental conditions." I am not sure this is a new method. In many other studies, authors used wake-model-based LUTs to estimate the efficiency of the wind farm. So the authors need to explicitly say what exactly is new.

***/ This whole paragraph has been revised to address your general comment to establish a better flow and provide more precise statements.

"Mittelmeier et al. (2013) presented a new method where not the absolute values between model and measurement are compared, but the relations between an observed turbine and all other turbines in the farm. In this way, the uncertainty of the measurement chain could be reduced. The model is also based on pre-calculated power matrices which we call from now on "lookup-tables" (LUTs). Different wake models or even combinations of wake model results can be used to provide results for these LUTs. But the model relies on measurements from a met mast which is often not available. Furthermore, with increasing size of wind farms, the assumptions of one measurement position being representative for the whole offshore wind farm is not valid (Dörenkämper, 2015).  Further investigations are necessary to obtain a reliable and automated method to detect underperformance at individual turbines in a wind farm."

15. Page 2 lines 27-28 "Especially: : : available." Already mentioned so remove it

***/ Sentence removed/***

16. Page 2 lines 31-33 Replace "this" by "the", add "of" after "method" and use "Mittelmeier et al. (2013)" instead of "(Mittelmeier et al., 2013). Replace "condition" by "conditions"

***/ New wording:
The purpose of this paper is to present the results of extending the wind farm performance monitoring method of Mittelmeier et al. (2013) by using SCADA instead of met mast data. A new combination of methods to obtain representative environmental conditions and further optimisation potential for wake models fine-tuned by SCADA data is presented and an estimation of the uncertainty of these methods is given./***

17. Page 3 line 1-2 "Hence the presented: : : LiDAR". Based on what you have already mentioned one can inferred what is written here so it is not necessary

***/ sentence is deleted/***

18. Page 3 line 4 it should be change to ": : :.of the method by Mittelmeier et al. (2013) is recalled".

***/ has been changed/***

19. Page 3 line 11 I know what you mean by "deviation" but you need to be exact so change to "A deviation between Pnpi and Pnmu"

***/ Thanks for this advice. It is actually the deviation between $\pi$ and $\mu$, that indicates underperformance. We have changes the first paragraph and hope this adds clarity:
"To detect underperformance of a wind turbine, we estimate the expected turbine power ratio $\pi$ (predicted power ratio) between the observed turbine and a reference turbine with a wake model for the actual condition and compare its result with the actual measured power ratio $\mu$. A deviation between $\pi$ and $\mu$, higher than a certain threshold indicates underperformance."/***

20. Page 3 lines 14-15 "The accuracy and calculation : : :. real wind farm flow" This is not true. A simple wake model can be as accurate as a complex one.

***/ True! We have revised the whole paragraph:
"The performance monitoring model (Fig. 1) is based on two dimensional LUTs. The user can choose any wake model or even a combination of different model results to provide power output $P_{\pi_{i,j}}$ values for different wind speed bin $i$ and wind direction bin $j$. The predicted power output $P_\pi$ is derived from the LUTs with linear interpolation knowing the measured wind speed and wind direction. "/***

21. Page 3 line 16 First "improve the underperformance detection capabilities" This is not always true. And replace "," by ";" before "on the other hand"

***/ Please see response given to Comment 20 /***

22. Page 3 line 23 Remove the first "Additional"

***/ is removed/***

23. Page 3 line 25 "for this purpose" I know what you mean but you have not mention any purpose and you want to refer to the monitoring method, I guess. So be precise

***/ "purpose" replaced by "monitoring method" /***

24. Page 3 line 28 Replace ": : : averages of 10 minutes" by ": : : averages over 10-min periods"

***/ has been replaced /***

25. Page 3 line 29 Replace ": : : averaging N quantities of 10 minutes time samples" by "averaging a number N of 10-min samples"

***/ has been replaced /***

26. Page 3 line 30 Remove "the averaging"

***/ "the averaging" has been removed /***

27. Page 4 line 1 You are talking about "correlation" but this is not a correlation of power it is a only a normalization. So as this is wrong the part of "This leads" does not make sense

***/ "correlated to " replaced by "divided by" /***

28. Page 4 line 4 "This is described in Eq. (1) and (2)" Well this is not described in the equations; the equations are simply the definition you are using for the normalized powers

***/ "This is described in Eq. (1) and Eq.(2)." Replaced by "We define"
After Eq. (1) "," replaced by "and". (from General Comment: Equations should not disrupt the text) /***

29. Page 4 line 12 Replace "can be described with Eq. (3)" by "is defined as"

***/ "can be described with Eq. (3):" replaced by "is defined as" /***

30. Page 4 lines 16-17 "where neta_ob,ref: : : (ob) turbine" You already defined everything so there is no need for this

***/ sentence deleted /***

31. Page 4 lines 26-29 So why do you have to use all the wind vanes (this is what is read from the text)? They could all have a different misalignment and so you will need to analyze each of them (in terms of wake deficits) if you want to use all of them.

***/ In fact, all wind direction signals are corrected for a certain bias, which may results from a combination of systematic yaw error and wrong north marking. We have used only one referencing (based on wake deficits) to conserve effects like the mentioned "wake drift". We want to use all wind vanes to cover the full variance which we use in our uncertainty calculation.
/***

32. Page 4 line 29 "complex area" what do you mean by complex area?

***/ "area" replaced by "plane" /***

33. Page 5 line 1 "of the scale" what do you mean by scale?

***/ "scale" replaced by "value range" /***

34. Page 5 line 2 "+-1.5IQR" be explicit. If the outliers removed are those outside the range +-1.5IQR then say so

***/ "defined by the" replaced by "outside"
The dot has been removed in the formula (General comment) /***

35. In Eq. 4 you have constants without units. If alpha is in degrees all these constants have the units of degrees and you need to state that

***/ Decimal separator changed from "," to "."
Product symbols removed (general comment), "Equation (4)" removed, so that the equation does not interrupt the text (general comment).
New wording:
"$\alpha = 1.3 \arctan\left(2.5 \frac{D_n}{L_n} + 0.15\right) + 10$ (4)
is proposed by IEC 61400-12-1 (2005) and defines the width of the disturbed sector in degrees seen by the downwind turbine (the constants have the dimension of degree). $D_n$ is the rotor diameter of the upwind turbine and $L_n$ the distance between the two turbines defined by Eq. **Fehler! Verweisquelle konnte nicht gefunden werden.**." /***

**36. Eq. 6 is not needed**

***/ the two sentences before Eq.6 and Eq.6 are deleted.
New wording:
"With $\beta$, being the angle between the wake inducing turbine and the northing and the wind direction $\vartheta$ the turbine wake indicator $\gamma$ can be described as:" /***

**37. Page 6 line 1 "the north inconsistency need different conditions" Yes obviously**

***/ sentence deleted /***

**38. Page 7 lines 1-2 "In Mittelmeier et al. (2015): : : wind speeds". Well that depends on the stability conditions. This will be true if compare unstable conditions with low wind speeds and high sigmas with neutral conditions with lower sigmas and higher wind speeds. But stable conditions will be in the low wind speed range with lower sigmas compared to neutral**

***/ You are right. In our demonstration wind farm, we have unfortunately no measurements which would allow us to calculate stability by Monin-Obukhov length. But we definitively do see a strong correlation with the turbulence intensity. We see higher turbulences at low wind speeds and lower turbulences at higher wind speeds. We have changed the wording, to be more precise:
"In Mittelmeier et al. (2015), we could show, that for the prevailing conditions at Ormonde wind farm $\sigma_a$ is a function of wind speed, decreasing with higher wind speeds." /***

**39. Page 7 line 3 "no impact" you mean "little impact"**

***/ "no impact" replaced by "little impact" /***

**40. Page 7 lines 7-8. This seems to be quite important and you do not provide any details about the study of Vollmer et al.**

***/ Thanks for pointing this out. We have now put more focus on explaining this parameter and a possible theory behind this observed wake drift.
The last paragraph of section 2.2 has been revised as follow:
"The third tuning parameter is applying a simple offset on the wind direction of the LUTs to account for a drift of the wake. We call this phenomena from here on "wake drift". Fleming (2013) has studied the effects of active wake control and in his baseline  simulation (no yaw error) a small wake drift to the right can be observed when looking downwind. In the LES study of Vollmer et al. (2016) the wake drift increases from neutral to stable conditions also for 0° yaw angle. Gebraad (2014, p86) gives an explanation for the observations from the simulations by Fleming (2013). The flow reacting on the rotation of the rotor causes the wake to rotate counter clockwise (looking downstream). Higher wind speeds from the upper layer are transported downwards (on the left side) and lower wind speeds from the lower layer are pushed upward on the right side of the wake. As a result the velocity deficit at the right part of the wake increases, so the wake deflects to the right.
Marathe et al. (2015) could show in their field measurement campaign with a dual-doppler radar the wake drifting to the right, as expected by the theory. But in the far wake they registered a movement to the left. The authors state the hypothesis that this contradicting phenomenon may be caused by atmospheric streaks. In an offshore field experiment by Beck et al. (2015) further evidence is provided that wakes are moving out of the centre line. This wake drift is currently not modelled in Fuga and therefore applied in a further step of the pre-process (Fig. 1)."

**References for this paragraph:**
Beck, H., Trujillo, J. J., Wolken-möhlmann, G., Gottschall, J., Schmidt, J., Peña, A., Gomes, V., Lange, B., Hasager, C. and Kühn, M.: Comparison of simulations of the far wake of alpha ventus against ship-based LiDAR measurements, in RAVE Conference., 2015.

Fleming, P. A., Gebraad, P. M. O., Lee, S., van Wingerden, J. W., Johnson, K., Churchfield, M., Michalakes, J., Spalart, P. and Moriarty, P.: Evaluating techniques for redirecting turbine wakes using SOWFA, in ICOWES2013 Conference., 2013.

Gebraad, P. M. O.: Data-Driven Wind Plant Control, 2014.

Marathe, N., Swift, A., Hirth, B., Walker, R. and Schroeder, J.: Characterizing power performance and wake of a wind turbine under yaw and blade pitch, , doi:10.1002/we, 2015.

Vollmer, L., Steinfeld, G., Heinemann, D. and Kühn, M.: Estimating the wake deflection downstream of a wind turbine in different atmospheric stabilities: An LES study, Wind Energy Sci. Discuss., (March), 1–23, doi:10.5194/wes-2016-4, 2016.

/***

41. Page 7 line 31 "usually about 4 to 6%" you need to give a reference here; otherwise show an example

***/ Unfortunately we have no reference for our own experience. Therefore we have changed the sentence and provide the following reference.
New wording:
"Results from the Offshore Wind Accelerator (OWA) (Clerc et al., 2016) provide a range of 2.5% to 5% combined uncertainty for offshore power curve verification based on a measurement chain that include a met mast and all its devices. The usage of LiDAR extends the range up to approximately 7%."

**Reference:**
Clerc, A., Stuart, P., Cameron, L., Feeney, S. and Fnc, I. C.: Results from the Offshore Wind Accelerator ( OWA ) Power Curve Validation using LiDAR Project, in Wind Europe Workshop "Analysis of operating wind farms.", 2016.

/***

42. Page 8 line 19 "is around 7%" based on what?

***/New wording:
"The uncertainty derived by Eq. **Fehler! Verweisquelle konnte nicht gefunden werden.**) can be displayed as a bandwidth around the underperformance indicator $\eta$, visualized in Fig. 4. Its magnitude is dependent on the sample size $N$. In Fig. 4 we obtain approximately 7 % uncertainty on the performance ratio for $N > 1000$. /***

43. Page 9 line 3 Figure 5: please show a proper layout with north orientation and Scales

***/ New Figure with orientation and scales provided

[Figure]

/***

44. Page 9 line 4 "6.3D" this is between turbines in the same row (does not look like that), between rows? And important what are rows for you: the rows of turbines in a particular direction or all "rows" of turbines?

***/ Not precise! You are right. Following text will add clarity:
"The farm layout displayed in Fig. 5 is structured in a regular array which allows comparison of several wake situations. The closest turbine spacing is in the range of 4.1 D to 4.3 D along the four rows orientated from north west to south east. We have selected a more frequent wind direction from south south west where multiple columns of four turbines are aligned with a distance ranging from 6.3 D to 6.5 D. To simplify the demonstration of underperformance detection we focused on single wake, double wake and triple wake conditions behind turbine OR26 for a south south westerly wind direction and a sector of 30° around the full wake situation. " /***

45. Page 9 line 7 Two years of "10-min" SCADA data?

***/ "for the following demonstration"  replaced by "to set up the performance monitoring model" /***

46. Page 9 line 10 "The quality of the derived wind direction" It is not the quality what you show there

***/ Ok, we have revised the wording as follow:
"The wind direction is supposed to be representative for the wind turbines in the monitoring model. In our example, we have averaged up to 30 corrected wind direction signals for each 10-min interval. The variation among the individual signals provides an uncertainty estimate for this artificial wind direction. In Fig. 6, a histogram of the full data set of two years with each count being the difference between a single vane measurement and the corresponding mean wind direction for the averaged period is visualized. This variation can nicely be described by a Gaussian distribution with standard deviation of 3.6°. This value is used for the uncertainty of the wind direction Table 1 is referring to." /***

47. Page 9 line 16 "from Section: : :" Explicitly state which equations

***/ there is not one explicit equation, it's the methodology we want to refer to.
"equations from" replaced by "methodologies described in" /***

48. Page 9 line 18 "binned into 2 deg" does not look like that but more like 4 deg.

***/ You are right. The 2 deg has been used for the model results and the SCADA results have been displayed with only 4 deg. We have changed the plot. It is now showing 180 instead of 90 points for the SCADA data /***

49. Page 9 line 23 These correlations are very high but the outliers seem to be also quite large so I am very skeptical about these computations
***/ The numbers where related to the correlations between modelled waked wind speed and the wind speed of the virtual met mast. This was in contradiction with the text. We have corrected the sentence:
"A linear regression between the wind speed of the standard model results  (red dashed line) and the SCADA measurements equals $R^2 = 0.96$. The improved model (green solid line) gives an $R^2 = 0.97$."/***

50. Page 9 The wake model seems to have a systematic bias when compared to the measurements (clearly seen at the highest deficits). Why? Is the average wind direction perhaps wrong? Perhaps you should average the wind direction of the turbines in the rows where the flow is not disturbed if you want to compare it with the wake model

***/ The wake model results used in this plot are directly taken from the Fuga Output. No wake model tuning/corrections as proposed in the pre-process step of the monitoring method (Fig. 1) has been applied.  The corrected model has an 2.5° offset for all single wake case, 3.5° offset for the double wake cases and a 4.5° offset for the triple wake case for wind directions 207° ±15°.  In Fig.7 the offset of the wind direction at 207° (the four demonstration turbines in a column) is approximately 2.2°. At the wind directions 132° and 312° , with the largest wake effects along the four rows of  7 to 8 turbines, the offset is approximately 5°. This fact supports the theory, that with every additional wake added to the flow, the overall "wake drift" increases. We will add this information into the caption of the plot.

"Figure 7: Wind farm averaged wind speed with wake effects normalised with wind farm averaged wind speed without wake effects plotted versus averaged wind farm wind direction. Black dots show the measurements from SCADA and the green solid line represents the results from Fuga with a Gauss averaging for standard deviation of 4°. An offset of the wind direction between model and SCADA can be observed. At 207° the offset is approximately 2.2° and it increases up to 5° for wind directions (132° and 312°) with the largest wake effects. An explanation and correction for this "wake drift" is proposed in section 2.2." /***

**51. Page 10 line 1 The wake model should be presented before the results in 3.1.2!**

***/ The wake model calibration is based on the "virtual met mast". This information was missing and will be provided in the revised version. Therefor we think it's helpful to read first about the wind speed and wind direction handling and then about the wake model calibration./***

**52. Page 10 line 11 How much data you use for the calibration period?**

***/ In our case, we have used two full years of SCADA data to obtain the plots in figure 9. In the current version of the monitoring model, no variation of stability is considered. Therefore the calibrated model should represent the annual average as good as possible. Small improvements could be expected, when extending the dimensions of the LUTs. But this is ongoing work and we think with the presented method we already achieved an underperformance detection level which is acceptable and helpful.
We will add the following clarification:
"The wake model is supposed to provide a two dimensional LUT (wind direction, wind speed) for each turbine. Further dimensions such as stability may improve the accuracy, but research and validation for these models are still ongoing. Therefore our calibrated model has to be representative for the average annual conditions. Two full years of SCADA data have been used for this task." /***

**53. Page 10 starboard and port terms are terms conventionally used in wake studies?**

***/ Sorry, being offshore tempt us to use nautical terms.
"Starboard" replaced by "right"
"Port" replaced by "left" /***

**54. Page 10 line 17 what do you mean by "global"**

***/ "global wind direction" replaced by "artificial wind direction from the virtual met mast (Section 2.1.1)" /***

**55. Page 10 lines 16-22 If this phenomenon occurs in 1 single wake the a very plausible reason is simply yaw misalignment**

***/ We will add the following wording:
**"**We cannot fully rule out the possibility of an unwanted yaw misalignment as the uncertainties within this process of aligning the turbine lies within 3° (IEC 61400-12-2, 2013). But a single wake drift of 2.5° is also within the simulation results for 0° yaw misalignment at stable conditions. (Vollmer et al., 2016)"

Further details to this topic are also given in the answers to the general comments.

References:

IEC 61400-12-2: Power performance of electricity-producing wind turbines based on nacelle anemometry., 2013.

Vollmer, L., Steinfeld, G., Heinemann, D. and Kühn, M.: Estimating the wake deflection downstream of a wind turbine in different atmospheric stabilities: An LES study, Wind Energy Sci. Discuss., (March), 1–23, doi:10.5194/wes-2016-4, 2016.

/***

**56. Page 10 line 32 "this data has been filtered for a wind direction sector of 5 deg"**

Change to "These data have been.."

***/ This data has" replaced by "these data have" /***

57. Page 11 line 2 You mean for prediction of what? AEP? Wind speed? A particular case?

***/ We have referenced all available power data of that specific case with the base model and with the tuned model output.
We have revised the sentence:
"The three fine-tuning steps decreased the power prediction error  in a full wake with ±5° sector width from 7% to 1.5%(Mittelmeier et al., 2015) for the presented case." /***

58. Page 11 line 5 Replace "has been" by "will be"

***/ "has been" replaced by "will be" /***

59. Page 11 line 6 what do you mean by "real data"? so before the data was not real?

***/ "real data and" has been removed /***

60. Page 11 line 9 degradation of 8% in terms of what?

***/ After ": :: 8% "  we have added  "of its power production " " : ::" /***

61. Page 11 line 13 Replace "in displayed" by "is shown"

***/ "in displayed" replaced by "is shown" /***

62. Page 11 line 18 Why "Therefore"?

***/ "Working with ::: wind speeds. Therefore ::: 5 m/s." is replaced by:
"Below 5m/s we have realized a strong increase in wind direction variation among the turbines compared to the artificial wind direction from the virtual met mast. This variation increases the uncertainty of the model and therefore its filtered out." /***

63. Page 12 line 10 "horizontal graph" you mean horizontal line?

***/ "graph" replaced by "line"

64. Symbols in Fig. 1 are not the same symbols: : : they are not in italics

***/ Fig. 1 has been improved. It is now also showing the pre-process of cleaning and applying offset corrections to the data, before wind speed and wind direction are derived.

[Figure]

/***

65. Figure 3: The wind direction should not be perpendicular to the rotor? nbeta (which is not the wind direction) is the angle perpendicular to the rotor

***/ $\beta$ is perpendicular to the rotor. The wind direction $\vartheta$ is in most cases different to $\beta$. And for this reason, we decided to display $\vartheta$ not perpendicular to the rotor. /***

66. Figure 5: scales, north, coordinates!

***/ Scales, location, and north have been added. Description and caption adjusted accordingly /***

67. Figure 10: there should be some green points below 0.6, so perhaps it is better to degrade based on the best Cp curve

***/ The green points below 0.6 are covered by the red points. We have changed the order of colour plotting. Now the points below the curtailment are green. The degradation is visible for all wind speeds /***

In addition to these comments we would like to point out, that by rechecking the filtering procedure a minor filtering bug was revealed (1.5 IQR was static and not dynamic). This has changed the uncertainty of the wind direction from 3.1° to 3.6° which leads to minor changes in the numbers given in Table 3 and Table 4.

---

## Author Response (AR1)

Paper: wes-2016-16
Title: Monitoring offshore wind farm power performance with SCADA data and advanced wake model
Authors: Niko Mittelmeier et al.

Answers to comments of anonymous Referee #1 by
Niko Mittelmeier et al. October 19, 2016

Dear Referee,

Thank you very much for your honest feedback and the very detailed and helpful comments. Your advices and suggestions will certainly help to make the paper clearer and better. In the first part below we have addressed each comment. Your overall message, "be more precise" is understood and we hope to meet your expectations. Our response is marked as follow: ***/ response /***. In the second part, we have attached the change track version of the word document with all referee comments. We hope this is helpful to find our (RC1) changes in the context. Please use the pdf bookmarks for navigation.

**Part 1**

**General comments**

I think that the subject in general is interesting as wind farm underperformance is an important issue that we sometimes do not want to discuss much in wind energy. Therefore, I started to read with interest the manuscript but at about the second page I became really bored of the continuous issues/typos/grammatical problems that the text has. It is not that the English is generally bad; it is more about the way the authors write sentences and connect the ideas. It is generally "very weird" the way they write. In the specific comments, I list a number of issues but as I said I became so bored of these things so I just did it for the first pages; in case the authors have the chance to resubmit, the manuscript has to pass many hands including some English technical experts before resubmitting.

***/ Introduction will be rewritten. Focus: "be more precise" (see comments below) /***

More important, the manuscript in its actual form reads more like a technical report describing a method rather than a scientific paper. The authors need to make clear what the contribution to science is (if any) and write the manuscript to establish that the method they suggest is clearly novel (so far I do not see the novelty; the wake model is not new, neither the uncertainty calculation). Also they make things harder to digest by their writing so the text needs some reshuffling to accomplish a good flow.

***/ For our wind farm monitoring model we have chosen the power matrix approached (look-up tables (LUTs)) which has been used in several studies before. (TC88 WG6, 2005 ), (Mellinghoff , 2007), (Carvalho and Guedes 2009) , (Westerhellweg et al., 2012) and (Mittelmeier et al. 2013).

We see the advantage of LUTs in the fact, that any wake model can be chosen to provide input for our model. And with further improved wake models, the monitoring method will improve.

The novelty is a new turbine referencing approach. Not the absolute values between model and measurement are compared, but the relation between an observed turbine and all other turbines in the farm. The uncertainty of the resulting performance ratio is much lower than the uncertainties of absolute production or AEPs. Furthermore all the above mentioned publications have proposed to use met mast data and we have demonstrated our method only with measurements which are available on state of the art wind turbines (SCADA data).

Usually measured data from nacelle mounted devices is prone to errors due to disturbed flow behind the rotor. When looking at absolute power values this would lead to high uncertainties. The IEC 61400-12-2 (2013) standard provides an example in Annex J showing uncertainties of approximately 20% on AEP for one turbine. A reduction in AEP uncertainty could be achieved by multiple measurements.

By using reference turbines, uncertainties from air density corrections can be neglected. Furthermore, the uncertainty of the wind speed has a much lower sensitivity factor. Wind direction measurements have a clear contribution to the combined uncertainties in our model, but we could show, that in our example, 7% uncertainty on the performance ratio are an improvement compared to existing methods.

The contribution to science is an explicit investigation on how underperformance can be detected in single wake, double wake and triple wake situations and we provide validation and suggestions how to improve results of the selected wake model in pre and post process. We have used Fuga because it is accessible, fast and easy to handle. But for experienced users of other models, the choice might be different and that's ok for our model.

**References:**

Carvalho, H. and Guedes, R.: Wind Farm Power Performance Test , in the scope of the 2 . Wind Farm Performance Matrix, , 1–9, 2009.

IEC 61400-12-2: Power performance of electricity-producing wind turbines based on nacelle anemometry., 2013.

Mellinghoff, H.: Wind Farm Power Performance Verification, DEWI., 2007.

Mittelmeier, N., Amelsberg, S., Blodau, T., Brand, A., Drueke, S., Kühn, M., Neumann, K. and Steinfeld, G.: Wind farm performance monitoring with advanced wake models, in Proceedings of the EuropeanWind Energy Association Offshore Conference, Frankfurt., 2013.

TC88 WG6, I.: Wind farm power performance testing working group draft, IEC., 2005.

Westerhellweg, A., Canadillas, B., Kinder, F., Neumann, T. and Windenergie-institut, D. G. D.: Farm Efficiency and Power Matrix based on RANS ( CFD ) Simulations for the offshore Wind Farm Alpha Ventus and Comparison with Measurements, 2012.

/***

About the subject: There is a clear shift of the direction of the wake even for the single wake case. The authors provide some arguments but in the single wake case the maximum wake deficit should simply be a 0 deg.

***/ We provide you with some recent publications which support the theory, that even in the wake of a single turbine with no yaw error a shift of the wake is observable. These studies have the general aim to investigate active wake control but they also provide examples for 0°yaw error. Fleming (2013) shows in his baseline simulation (no yaw error) a small wake shift to the right when looking downwind. In the LES study of Vollmer et al. (2016) it can be observed, that the wake deflection increases from neutral (Vollmer et al. 2016, Fig. 5)  to stable conditions(Vollmer et al. 2016, Fig. 9). These Figures provide simulated results also for a turbine with 0° yaw angle. In both cases the maximum wake deficit is found to be on the right side of the centre line (looking downstream).
Gebraad (2014, p86) gives an explanation for the observations from the simulations by Fleming (2013). The flow reacting on the rotation of the rotor causes the wake to rotate counter clockwise (looking downstream). Higher wind speeds from the upper layer are transported downwards (on the left side) and lower wind speeds from the lower layer are pushed upward on the right side of the wake. As a result the velocity deficit at the right part of the wake increases, so the wake deflects to the right.
Marathe et al. (2015) could show in their field measurement campaign with dual-doppler radar, that in the near wake region, the wake is drifting to the right, as expected by the theory. But in the far wake they registered a contradicting movement. The authors state the hypothesis that this phenomenon may be caused by atmospheric streaks. An offshore field experiment by Beck et al. (2015) provides further evidence that wakes are moving out of the centre line.

**References:**

Beck, H., Trujillo, J. J., Wolken-möhlmann, G., Gottschall, J., Schmidt, J., Peña, A., Gomes, V., Lange, B., Hasager, C. and Kühn, M.: Comparison of simulations of the far wake of alpha ventus against ship-based LiDAR measurements, in RAVE Conference., 2015.

Fleming, P. A., Gebraad, P. M. O., Lee, S., van Wingerden, J. W., Johnson, K., Churchfield, M., Michalakes, J., Spalart, P. and Moriarty, P.: Evaluating techniques for redirecting turbine wakes using SOWFA, in ICOWES2013 Conference., 2013.

Gebraad, P. M. O.: Data-Driven Wind Plant Control, 2014.

Marathe, N., Swift, A., Hirth, B., Walker, R. and Schroeder, J.: Characterizing power performance and wake of a wind turbine under yaw and blade pitch, , doi:10.1002/we, 2015.

Vollmer, L., Steinfeld, G., Heinemann, D. and Kühn, M.: Estimating the wake deflection downstream of a wind turbine in different atmospheric stabilities: An LES study, Wind Energy Sci. Discuss., (September), 1–23, doi:10.5194/wes-1-129-2016, 2016.

/***

The authors use a nacelle-based vane
for the wind direction so why not checking if there is a systematic turbine misalignment
by looking at the nacelle position signal in the SCADA data?

***/ For our monitoring model we are using the absolute wind direction signal from each turbine which is defined as

$$\vartheta = nacelle\ position + wind\ vane\ position$$

The nacelle position is the angle between the rotor axis and a marking for true north. This marking is calibrated as part of the commissioning. But often this signal is not maintained well during operation, because it has no effect on turbine performance. This causes the necessity to apply a bias correction to this signal before using it for reanalysis purposes. The wind vane position indicates the angle of the flow to the rotor axis. It directly provides a value for the yaw error. The turbine controller uses this signal to control the yaw activity. For an infinite averaging the mean value of the wind vane position is 0°. We have used 12127 10-min values for the wake model calibration. A histogram of the vane position signal for the whole data is provided in Figure x below.

[Figure]

Figure x: Histogram of the wind vane position signal from turbine 26 showing all data that has been used for wake model calibration. Resulting in a mean value of -0.2 and a median of -0.3.

A systematic yaw error resulting from the sensor alignment can be estimated to ±2° for a single turbine (IEC 61400-12-2, 2013). A nacelle transfer function for wind direction is used to take the effects of the rotor into account. We do not assume this correction being perfect, so we end up with uncertainties of approximately 3°.

We will include these clarifications in Section 2.1.1. And add further explanations to the data handling, filtering and corrections that are necessary to obtain the right quality signal. (See also Comment 64)

**Reference:**
IEC 61400-12-2: Power performance of electricity-producing wind turbines based on nacelle anemometry., 2013.

/***

The authors also refer to
the study of Vollmer, but in that study, the wake is deflected intentionally by misaligning the turbines. So the most plausible explanation it is simple yaw misalignment unless the authors discard this by showing that the turbines are indeed not misaligned (but they do not do that).

***/ In the study of Vollmer, simulations are showing wake behaviour for 30°, 0° and -30° yaw angle under different atmospheric conditions. Even at 0° yaw angle the maximum wake deficit is not at the centre line (perpendicular to the rotor). This effect increases with increasing atmospheric stability.

Unfortunately we don't have information from the site which would allow a stability classification based on the Monin - Obukhov length. But we obtained the best wake deficit fit between the SCADA data and the Fuga calculations with a $\zeta_0 = 2.72\,e-7$ which is supposed to be used for more stable cases.

We have changed the reference to the following publication:
Vollmer, L., Steinfeld, G., Heinemann, D. and Kühn, M.: Estimating the wake deflection downstream of a wind turbine in different atmospheric stabilities: An LES study, Wind Energy Sci. Discuss., (September), 1–23, doi:10.5194/wes-1-129-2016, 2016.

/***

There is a general problem with the way the authors make references in the text and the reference list itself at the last section. You should write refs. in the text as: "A power curve is given for each turbine (Smith, 2001). However, Jonas (2010) described another method. Such method was also shown in some previous studies (Klinsmann, 2006; Pauli, 2010)". In the specific comments I select some specific cases but most of the references are wrongly made. And the references in the reference list should be made consistently: Names, title (non-capital all refs. or all capital), etc. Such type of reference list makes me wonder about the quality of the whole study. The reference list should be made with consistency.

Also you have a problem with the equations;
they are part of the text and should not disrupt it! The dot symbol does not mean multiplication, it means dot product but you don't have such products.

The "same" symbols are sometimes in italics and sometimes in normal text; if they are the symbol of the same thing then they should be written in the same way.

***/ Thanks for pointing this out. It will be improved accordingly. /***

**Specific comments:**

1. Page 1 line 16 ": : :technical solutions." This type of statements are very general and not precise and specific. What do you mean by this? Turbines, models, methods?

***/ The intention was to start the paper with a very generic statement cause not only good turbines will make an investment successful. Installation, O&M, grid components, models, monitoring methods and guarantees are also important. But we agree, being more precise here will help the reader.

We will change the wording from "technical solution" to "wind turbine" /***

2. Page 1 line 18 ": : :definition (: : :) defines: : :" that the system is ready to operate" this is redundant. Why not ": : :definition (: : :) is that related to a system ready to operate"

***/ New wording:
"In wind industry, the common standard IEC TS 61400-26-1 (2011) defines different categories of turbine conditions and describes the calculation of availability". /***

3. Page 1 line 20 ": : : quality and quantity." Of what? In the next paragraph you kind of explain it but you cannot simply say this here and expect that the reader finds the answer later. If this is the case then that sentence can be removed.

***/ You are right. The sentence is moved to the next paragraph. /***

4. Page 1 line 21 Replace ": : :much SCADA" by "lots of" or "a good amount of"

***/ "much" replaced by "lots of" /***

5. Page 1 lines 27-28 "Work on: : : of 2016". You don't need this reference and does not help the paper so remove it

***/ Sentence is removed. /***

6. Page 1 lines 28- Page 2 line 1 "For most turbines: : :. wake effects" You make it sound as it was only a problem for offshore turbines and it is not so replace by e.g. "For most turbines in a typical wind farm, verification of the performance by comparison with the power curve is not suitable: : :" then ": : : maintenance of a met mast is very expensive particularly offshore."

***/ New wording:
"For most turbines in a typical wind farm, verification of the performance by comparison with the power curve is not suitable due to wake effects. And the installation and maintenance of a met mast is very expensive particularly offshore." /***

7. Page 2 line 3 Replace "accounts" by "account"

***/ replaced /***

8. Page 2 line 6 "Incorrect parameter settings" you mean "turbine parameter"?

***/ New wording: "Incorrect turbine parameter settings…" /***

9. Page 2 line 9 "turbine has a limited power output which has been externally applied" You want to refer to the limit but with the use of the "which" you mean the limited power output but surely this is not what is externally applied because one cannot apply a limited power output: : : that is a consequence of limiting something else.

***/ New wording:

"A curtailed turbine has a limited power output below its expected power. Possible reasons for curtailments are load reduction or grid requirements. For these incidents, turbine parameters are changed on purpose and therefore documented in the turbines SCADA logs." /***

10. Page 2 lines 11-16 this is a very weird paragraph. "Upwind turbines influencing the free flow for downwind turbines". This is a weird sentence because the flow is not free for the downwind turbines. Why not just removing the "free" word. Then it is also weird because you have ": : :, wake effects," so with this construction you say the wake effects are turbines! Then you have these references to Albers (all wrongly made; see my major comment). Then you say "Albers has also looked: : :flow models": this is a personal communication or is in one of his studies? Then comes "But at that time: : :" what time? Which year or which study in particular? There is also a "to be further development" that should be "to be further developed"

***/ The whole paragraph has been revised to be more precise:
"Albers (2004) has published two methodologies for wind turbine performance evaluation. His integral model uses available wind conditions from the energy production of neighbouring WTs, met masts or a combination of both and transfers the information via flow modelling and wake modelling to the investigated wind farm. The measured yield is corrected for turbine availability and then compared against the modelled yield in absolute values. Due to high uncertainties this method is only proposed as a first general check. To reveal smaller deviations he proposes a relative wind turbine performance evaluation model.  For this method, active power of direct neighbours are plotted against each other and by comparing two periods, changes can be evaluated. This method explicitly excludes the sectors where wakes are effecting one of the two turbines."

**Part 1**

It is an interesting paper, introducing a new validation method for identifying wind farm underproduction. Such methods are highly needed with the large amount of wind turbines are installed in wind farms. The precondition for my review is that the method should also be applicable for implementation and not only be an theoretical exercise. The method, which seems to be a spin-off from an EERA project named ClusterDesign, refers to an ideal determination of the inflow conditions. The proposed method uses wake models estimates as reference, which seems to make a robust estimate of the underproduction. The accumulated uncertainty for the inflow conditions has been estimated to 7% and this number seems realistic when using recent calibrated instruments (cup and vane). This number is not realistic when using derived inflow conditions based on nacelle anemometry, electric power and wind turbine yaw position for periods longer than 1 year according to my experience.

***/ You are right, this result would not be realistic if it was based on absolute AEP values. But we have obtained 7% combined uncertainty based on normalised reference values compared between model and measurement. We see the advantage of the proposed method in the fact, that a precise wind speed measurement is less important compared to the IEC 61400-12-1 or IEC 61400-12-2. The sensitivity factor for wind speed uncertainty is taken from the slop of the power curve. When we normalize the power curve with the power of the same turbine type, Type B uncertainties become almost 0 in free flow conditions.  The second thing we find helpful to reduce the know disadvantages from nacelle  wind speed measurements is the fact, that we average all devices from turbines in free flow conditions. The variation among these signals is then represented in the uncertainty estimation.

We can confirm your concern, that yaw positions (nacelle positions) are prone to errors over time if no care is taken. For our monitoring model we are using the absolute wind direction signal from each turbine which is defined as

$$\vartheta = nacelle\ position + wind\ vane\ position$$

The nacelle position is the angle between the rotor axis and a marking for true north. This marking is calibrated as part of the commissioning. But often this signal is not maintained well during operation, because it has no effect on turbine performance. This causes the necessity to apply a bias correction to this signal before using it for reanalysis purposes. The wind vane position indicates the angle of the flow to the rotor axis. It directly provides a value for the yaw error. The turbine controller  uses this signal to control the yaw activity.  We have recalibrated $\vartheta$ by looking at the maximum wake deficit behind the turbine 26. This offset has been used to recalibrate all nacelle positions in the farm. /***

Problem: The determination of the wind farm inflow (environmental) conditions (wind speed and wind direction) seems not to be aligned with the state-of-art wind farm signals.

***/ One of our main objectives was to establish a method that can be applied with no need for additional hardware installations. Therefor we have used the available SCADA data and a pre-process to derive the wind farm inflow conditions. We agree, with LiDAR techniques improvements may be possible./***

In section 2.1.1 the wind direction is derived, but without any reference to how this is done. The wind direction measured on the nacelle is only used for yaw control, where the strategy is the keep the rotor aligned with the wind direction to minimize the yaw-misalignment. This signal can also identify a "forced" yaw misalignment used to determine the "wake drift"? The optimal readings from this instrument is 0, and will not reveal anything about the actual flow direction, which only can be identified from the wind turbine yaw position. The wind turbine yaw position not used by the controller, only when wind farm has sector management (proposed but never seen). The yaw position is usually not calibrated or has a wrong offset, which need to be identified.

***/ Thanks for pointing this out. We see the need to provide more clarification in this section. We will add the following explanations:

"The first step is to derive a wind direction $\vartheta$ for each 10 min interval. For our monitoring model we are using the absolute wind direction signal from each turbine which is defined as

$$\vartheta = nacelle\ position + wind\ vane\ position \qquad\qquad\qquad (4)$$

The nacelle position is the angle between the rotor axis and a marking for true north. This marking is calibrated as part of the commissioning. But often this signal is not maintained well during operation, because it has no effect on turbine performance. This causes the necessity to apply an offset correction to this signal before using it for reanalysis purposes. The wind vane position indicates the angle of the flow to the rotor axis. It directly provides a value for the yaw error. The turbine controller uses this signal to control the yaw activity. Within the Pre-Process (Fig.1) of the monitoring model we estimate the north marking offset for one turbine by checking the location of the maximum wake deficit with respect to the true north. Then we compare the average wind direction between corrected turbines and neighbouring turbines to estimate the remaining offset for all turbines. After applying this offset correction, the wind direction from all wind vanes are averaged in the complex plane to account for the wind direction discontinuity at the beginning/end of the value range, after removing outliers outside ±1.5 IQR (interquartile range)."

Furthermore, we will update Fig. 1 as below, to show the Pre-Process which is necessary to derive the right wind direction signal.

[Figure]

/***

Section 2.1.2 states to use nacelle anemometry to determine the wind speed; correct this is the only accessible wind speed measured on a wind turbine. This is measured with either a cup anemometer or sonic, located on the nacelle (behind the rotor). The signal is recorded through the controller and stored in SCADA system, but lacks documentation and uncertainty estimation. A correlation check between a number of identical wind turbines reveals a large scatter in the binned power curves. The scatter increases when the turbine operates in a wake compared to free inflow.
Conclusion:
the nacelle wind speed signal is biased. Furthermore the nacelle anemometer changes over years e.g. due to degradation. Even a NTF based wind speed (IEC 61400-12-2) is only applicable for free, undisturbed inflow.

***/ We absolutely agree with you conclusion. But as described in our response above, the monitoring method is less sensitive to wind speed measurements compared to IEC 61400-12-1 and IEC 61400-12-2 cause comparison is based on normalized power curves.
Secondly, the derived wind speed consists of only free and undisturbed nacelle wind speed signals and the variation among the devices is reflected by the uncertainty (one standard deviation) /***

Conclusion on inflow conditions: the stated uncertainty, for wind speed and wind direction does not meet the requirements given in IEC 61400-12-1 and this need to be addressed both in the method and in the example.

***/ We will add clarification about that in the last paragraph of Section 4 (Discussion):
"The stated uncertainties for wind speed and wind direction may be sufficient for the relative comparison to detect underperformance between turbines but it does not meet the requirements for an absolute performance validation according to IEC 61400-12-1(2005) or IEC 61400-12-2 (2013). One could perform power curve verification test in accordance with the mentioned standards at turbines where its applicable and those turbines being reference turbines in the monitoring method would increase the confidence in underperformance detection. At least for the concurrent period."
/***

Comments to the figures: all figures should include proper captions readable out of context. The caption of the figures are not sufficient e.g. while Figure 2b is not a addressed in the caption.

***/ Thanks for this advice. Below we provide new captions for each figure so that its understandable out of the context:

[revised manuscript text omitted]

/***

**The description of the method seems to be adequate, but the "wake drift" in section 2.2 is not well defined, I assume this refers to periods with active wake control, which I do not expect has been implemented yet?**

***/ The data for our demonstration is without active wake control. We cannot fully rule out the possibility of an unwanted yaw misalignment as the uncertainties within this process of aligning the turbine lies within 3° (IEC 61400-12-2, 2013).

We rather think, that a small wake drift is also possible for a perfectly aligned turbine. Therefore we will add the following explanations and references to provide better explanations of the observed phenomenon.

"The third tuning parameter is applying a simple offset on the wind direction of the LUTs to account for a drift of the wake. We call this phenomena from here on "wake drift". Fleming (2013) has studied the effects of active wake control and in his baseline simulation (no yaw error) a small wake drift to the right can be observed when looking downwind. In the LES study of Vollmer et al. (2016) the wake drift increases from neutral to stable conditions also for 0° yaw angle. Gebraad (2014, p86) gives an explanation for the observations from the simulations by Fleming (2013). The flow reacting on the rotation of the rotor causes the wake to rotate counter clockwise (looking downstream). Higher wind speeds from the upper layer are transported downwards (on the left side) and lower wind speeds from the lower layer are pushed upward on the right side of the wake. As a result the velocity deficit at the right part of the wake increases, so the wake deflects to the right.
Marathe et al. (2015) could show in their field measurement campaign with a dual-doppler radar the wake drifting to the right, as expected by the theory. But in the far wake they registered a movement to the left. The authors state the hypothesis that this contradicting phenomenon may be caused by atmospheric streaks. In an offshore field experiment by Beck et al. (2015) further evidence is provided that wakes are moving out of the centre line. This wake drift is currently not modelled in Fuga and therefore applied in a further step of the pre-process (Fig. 1)."

**References for this paragraph:**
Beck, H., Trujillo, J. J., Wolken-möhlmann, G., Gottschall, J., Schmidt, J., Peña, A., Gomes, V., Lange, B., Hasager, C. and Kühn, M.: Comparison of simulations of the far wake of alpha ventus against ship-based LiDAR measurements, in RAVE Conference., 2015.

Fleming, P. A., Gebraad, P. M. O., Lee, S., van Wingerden, J. W., Johnson, K., Churchfield, M., Michalakes, J., Spalart, P. and Moriarty, P.: Evaluating techniques for redirecting turbine wakes using SOWFA, in ICOWES2013 Conference., 2013.

Gebraad, P. M. O.: Data-Driven Wind Plant Control, 2014.

Marathe, N., Swift, A., Hirth, B., Walker, R. and Schroeder, J.: Characterizing power performance and wake of a wind turbine under yaw and blade pitch, , doi:10.1002/we, 2015.

Vollmer, L., Steinfeld, G., Heinemann, D. and Kühn, M.: Estimating the wake deflection downstream of a wind turbine in different atmospheric stabilities: An LES study, Wind Energy Sci. Discuss., (March), 1–23, doi:10.5194/wes-2016-4, 2016.

/***

Paper: wes-2016-16
Title: Monitoring offshore wind farm power performance with SCADA data and advanced wake model
Authors: Niko Mittelmeier et al.

Answers to comments of anonymous Referee #3 by

Niko Mittelmeier et al. November 1, 2016

Dear Referee,

Thank you very much for your interest and taking the time to review our paper. Your comments and suggestions are very helpful and highly appreciated. In the first part, we have addressed each comment and our responses are marked as ***/ response/***. In the second part, we have attached the change track version of the word document with all referee comments. We hope this is helpful to find our changes (RC3) in the context. Please use the pdf bookmarks for navigation.

**Part 1**

Dear Authors,
thank you for a interesting and novel idea for the detection of underperforming wind turbines. While I agree with the overall tenet of the paper, there are a few issues I would like to have clarified. Particularly, the SCADA system delivers data at a much higher rate than the 10-min averages. What would happen if you'd make use of the 1-sec resolution available from the data? 130 values would suddenly be 2 minutes instead of 21 hours, if every assumption stays unchanged - which it probably doesn't.

***/That is a very good question. And you are right, SCADA data can be recorded with 1-sec resolution. For our particular case, we have used two years of measurements to obtain a fairly good average result for the tuned wake model. And for this period only 10-min data was available. We see the problem with the higher resolution in the increasing scatter of the measurement. Particularly for 1-sec data wind direction and wind speed measured behind the rotor has a very large variation. As we are using this variation to determine the uncertainty of the measurement we would obtain a much higher uncertainty for the detection of underperformance. Type A uncertainties will decease faster  but Type B uncertainties will increase.  /***

How do you deal with intra-10-min variability? Do you require a relatively stable weather situation or at least wind direction to be able to do it?

***/ We are working with 10-min averages and we need $N$ values of 10-min to be averaged, before we can highlight underperformance. Therefore we think that intra-10-min variability is averaged out in most cases. The intra-10-min variability of the wind direction is supposed to be covered by the Gaussian averaging method proposed by Gaumond et al. (2014).
The model is tuned for the annual average weather conditions. In weather conditions, where the model systematically under-, or over-predicts the wake effects the underperformance indicator $\eta$ will be biased. This has not been explicitly tested, but we think that n(n-1) calculations would reveal turbines which deviate compared to the fleet.
A degradation of the whole wind farm will probably be a slow process and would therefore be much closer at annual averaged conditions for which the wake model is tuned.

P3L17: I think this is debatable. Since you need quite a number of values / quite some time to detect the deviations, the method is not really real time anyway, so it could also be analysed retroactively every now and then. It also could be run on a larger highperformance
computer based on downloaded SCADA data. Besides, the connection between higher computational cost and accuracy of the wake models is sketchy at best, see e.g. the results presented in WindBench.

***/You are right, this whole paragraph has been revised as follow:
"The performance monitoring model (Fig. 1) is based on two dimensional LUTs. The user can choose any wake model or even a combination of different model results to provide power output $P_{\pi_{i,j}}$ values for different wind speed bin $i$ and wind direction bin $j$. The predicted power output $P_\pi$ is derived from the LUTs with linear interpolation knowing the measured wind speed and wind direction. "

With real time calculation speed, we meant the fact, that in our performance monitoring model each and every single 10-min SCADA measurement is recalculated by the wake model and therefore the wake model must be at least be able to calculate one single conditions in 10-min. The detection of underperformance is due to the averaging of many 10-min samples not "real time" anymore.
/***

2.1.1: I wonder why you do not use the same correction method on all the wind farms wind vanes? It just requires SCADA data and some computer power, so it would not be too difficult. Also, when you're calculating a mean wind direction for the whole farm, aren't you relying on a smaller size farm far away from the coast? For example, if you have a location like Anholt, then you have wind and direction gradients due to the proximity of the land across the wind farm. How would that influence your method and its accuracy?

***/This is a very valid point. In our model, the wind direction should be representative for the selected turbines. The variation among the individual wind vanes is used to come up with an uncertainty estimate. With larger wind farms, we expect this variation to increase which will lead to an increase of the uncertainties. We would recommend to divide very large wind farms in smaller groups. One way of defining a quality criteria could be to ensure, that the standard deviation of the single wind directions compared to the average wind direction is smaller or equal to 3.6° (values we have obtained).
We will add the following wording in the paper, to better describe the wind direction correction and estimation:

"The first step is to derive a wind direction ϑ for each 10 min interval. For our monitoring model we are using the absolute wind direction signal from each turbine which is defined as

$$\vartheta = nacelle\ position + wind\ vane\ position \tag{4}$$

The nacelle position is the angle between the rotor axis and a marking for true north. This marking is calibrated as part of the commissioning. But often this signal is not maintained well during operation, because it has no effect on turbine performance. This causes the necessity to apply an offset correction to this signal before using it for reanalysis purposes. The wind vane position indicates the angle of the flow to the rotor axis. It directly provides a value for the yaw error. The turbine controller  uses this signal to control the yaw activity.
Within the Pre-Process (Fig.1) of the monitoring model we estimate the north marking offset for one turbine by checking the location of the maximum wake deficit with respect to the true north. Then we compare the average wind direction between corrected turbines and neighbouring turbines to estimate the remaining offset for all turbines. After applying this offset correction, the wind direction from all wind vanes are averaged in the

complex plane to account for the wind direction discontinuity at the beginning/end of the value range, after removing outliers outside ±1.5 IQR (interquartile range)."

We are using the wake "centre check" only at one turbine because of the observed "wake drift" (We will give more explanation to this term three comments later) /***

P4L1: Is that one reference turbine for the whole farm, or one particular one for each of the other turbines? If it is one for the farm, how is it defined?

***/ we take one turbine (turbine under observation) and use every single turbine in the farm as reference. (One after each other) Then we take the next turbine to observe. In this way, we obtain n(n-1) results. In this way we increase the confidence in underperformance detection cause a real underperforming turbine will be highlighted multiple times. /***

P7L23: So the Type A uncertainties do not multiply in a multipe wake situation?

***/ It certainly makes a difference in the Type A uncertainties in which wake situation you are and this is reflected by the prediction accuracy of this wake itself. We use the standard deviation of $N$ differences between the modelled power and the measured power and divide it with the square root of $N$.

We have changed the wording to be more precise:
"For the predicted power $P_\pi$, we are using a combined uncertainty with statistical type A uncertainties, being the experimental standard deviation of the mean from the difference between wake model predictions and measurements and type B uncertainties which conclude from the instrument devices to estimate wind speed and wind direction. "/***

P10L15-22: This is quite interesting. Has this behaviour been observed anywhere else? Another explanation could be that the overall wind flow is skewed at the Ormonde location, which judging by the map is not impossible, seeing that the wind farm is wedged between land and a larger offshore wind farm. Or do I understand this wrong, and it is an effect from Fuga which is described here? Did you switch on the meandering mechanism in Fuga?

***/ The effect we are describing here is not modelled in Fuga. During our wake model results validation, we have also looked into the meandering option, but we obtained better results with the Gaussian averaging method. It might also be possible, that the neighbouring wind farms have an effect and play a role on the observed behaviour. But we can also provide some publications below which have observed and tried to explain a similar wake behaviour.

These studies have the general aim to investigate active wake control but they also provide examples for 0°yaw error. Fleming (2013) shows in his baseline simulation (no yaw error) a small wake shift to the right when looking downwind. In the LES study of Vollmer et al. (2016) it can be observed, that the wake deflection increases from neutral (Vollmer et al. 2016, Fig. 5) to stable conditions (Vollmer et al. 2016, Fig. 9). These Figures provide simulated results also for a turbine with 0° yaw angle. In both cases the maximum wake deficit is found to be on the right side of the centre line (looking downstream).
Gebraad (2014, p86) gives an explanation for the observations from the simulations by Fleming (2013). The flow reacting on the rotation of the rotor causes the wake to rotate counter clockwise (looking downstream). Higher wind speeds from the upper layer are transported downwards (on the left side) and lower wind speeds from the lower layer are pushed upward on the right side of the wake. As a result the velocity deficit at the right part of the wake increases, so the wake deflects to the right.
Marathe et al. (2015) could show in their field measurement campaign with dual-doppler radar, that in the near wake region, the wake is drifting to the right, as expected by the theory. But in the far wake they registered a contradicting movement. The authors state the hypothesis that this phenomenon may be caused by atmospheric streaks. An offshore field experiment by Beck et al. (2015) provides further evidence that wakes are moving out of the centre line.

/***

Figure 5: A map of the location would be good here (see above).

***/ New Figure with orientation and scales provided

[Figure]

/***

Figure 7: There seems to be a shift in wind direction between the SCADA system and the calculations - any idea where that is coming from?

***/ The wake model results used in this plot are directly taken from the Fuga Output. No wake model tuning/corrections as proposed in the pre-process step of the monitoring method (Fig. 1) has been applied. The corrected model has an 2.5° offset for all single wake case, 3.5° offset for the double wake cases and a 4.5° offset for the triple wake case for wind directions 207° ±15°. In Fig.7 the offset of the wind direction at 207° (the four demonstration turbines in a column) is approximately 2.2°. At the wind directions 132° and 312° , with the largest wake effects along the four rows of 7 to 8 turbines, the offset is approximately 5°. This fact supports the theory, that with every additional wake added to the flow, the overall "wake drift" increases. We will add this information into the caption of the plot.

"Figure 7: Wind farm averaged wind speed with wake effects normalised with wind farm averaged wind speed without wake effects plotted versus averaged wind farm wind direction. Black dots show the measurements from SCADA and the green solid line represents the results from Fuga with a Gauss averaging for standard deviation of 4°. An offset of the wind direction between model and SCADA can be observed. At 207° the offset is

approximately 2.2° and it increases up to 5° for wind directions (132° and 312°) with the largest wake effects. An explanation and correction for this "wake drift" is proposed in section 2.2." /***

Table 2: Is the source of that the IEC Annex D uncertainty estimation, or own values?

***/The values are based on the example given in IEC 61400-12-1 (2005) Annex E. We will add this information in the caption of Table 2./***

**Textual issues:**
P1L9: The presented method or the present method?

***/We will change the sentence to: " The method, presented in this paper, estimates…/***

P1L18: "The common [...] definition defines that the system is ready to operate" might not be exactly what is in the availability standard. Please rephrase.

***/Thanks for the advice, here is our new sentence:
"In wind industry, the common standard IEC TS 61400-26-1 (2011) defines different categories of turbine conditions and describes the calculation of availability"/***

P1L23: What's the difference between IEC 61400 and IEC TS 61400?

***/ TS stands for "Technical Specification".
Both Documents (IEC TS 61400-26-1 and IEC TS 61400-26-2) are technical specifications. An "International Standard" has a higher degree than a "technical specification". Work is ongoing to bring these TS to the level of an "International Standard"
We have revised the references accordingly./***

P2L25: To lower uncertainties, ... ??

***/We have rephrase the whole paragraph:
"Mittelmeier et al. (2013) presented a new method where not the absolute values between model and measurement are compared, but the relations between an observed turbine and all other turbines in the farm. In this way, the uncertainty of the measurement chain could be reduced. The model is also based on pre-calculated power matrices which we call from now on "lookup-tables" (LUTs). Different wake models or even combinations of wake model results can be used to provide results for these LUTs. But the model relies on measurements from a met mast which is often not available. Furthermore, with increasing size of wind farms, the assumptions of one measurement position being representative for the whole offshore wind farm is not valid (Dörenkämper, 2015). Further investigations are necessary to obtain a reliable and automated method to detect underperformance at individual turbines in a wind farm."/***

P6L15: You might want to explain what you mean by "wake drift".

***/Sure. We understand the need to provide more explanation and references on this topic. The last paragraph of section 2.2 has been revised as follow:
"The third tuning parameter is applying a simple offset on the wind direction of the LUTs to account for a drift of the wake. We call this phenomena from here on "wake drift". Fleming (2013) has studied the effects of active wake control and in his baseline simulation (no yaw error) a small wake drift to the right can be observed when looking downwind. In the LES study of Vollmer et al. (2016) the wake drift increases from neutral to stable conditions also for 0° yaw angle. Gebraad (2014, p86) gives an explanation for the observations from the simulations by Fleming (2013). The flow reacting on the rotation of the rotor causes the wake to rotate counter clockwise (looking downstream). Higher wind speeds from the upper layer are transported downwards (on the left side) and lower wind speeds from the lower layer are pushed upward on the right side of the wake. As a result the velocity deficit at the right part of the wake increases, so the wake deflects to the right.
Marathe et al. (2015) could show in their field measurement campaign with a dual-doppler radar the wake drifting to the right, as expected by the theory. But in the far wake they registered a movement to the left. The authors state the hypothesis that this contradicting phenomenon may be caused by atmospheric streaks. In an offshore field experiment by Beck et al. (2015) further evidence is provided that wakes are moving out of the

centre line. This wake drift is currently not modelled in Fuga and therefore applied in a further step of the pre-process (Fig. 1)."

**References for this paragraph:**

Beck, H., Trujillo, J. J., Wolken-möhlmann, G., Gottschall, J., Schmidt, J., Peña, A., Gomes, V., Lange, B., Hasager, C. and Kühn, M.: Comparison of simulations of the far wake of alpha ventus against ship-based LiDAR measurements, in RAVE Conference., 2015.

Fleming, P. A., Gebraad, P. M. O., Lee, S., van Wingerden, J. W., Johnson, K., Churchfield, M., Michalakes, J., Spalart, P. and Moriarty, P.: Evaluating techniques for redirecting turbine wakes using SOWFA, in ICOWES2013 Conference., 2013.

Gebraad, P. M. O.: Data-Driven Wind Plant Control, 2014.

Marathe, N., Swift, A., Hirth, B., Walker, R. and Schroeder, J.: Characterizing power performance and wake of a wind turbine under yaw and blade pitch, , doi:10.1002/we, 2015.

Vollmer, L., Steinfeld, G., Heinemann, D. and Kühn, M.: Estimating the wake deflection downstream of a wind turbine in different atmospheric stabilities: An LES study, Wind Energy Sci. Discuss., (March), 1–23, doi:10.5194/wes-2016-4, 2016.

/***

P9L9/L15: "of demonstration wind farm" could be deleted without detriment.

***/ Thanks for pointing this out. We will delete that part. /***

P9L13: This value is used_for the uncertainty...

***/ Thanks. It is corrected./***

P10L5: I would drop the brackets around the version number of Fuga.

***/ Agreed./***

P10L14: I don't think I've seen zeta_naught introduced before?

***/ This value is directly used as an input parameter in Fuga. The theory behind it is described in Ott and Nielsen (2014). We will add "Secondly, the **Fuga** parameter to model the effect of atmospheric stability $\zeta_0$…"

References:
Ott, S. and Nielsen, M.: Developments of the offshore wind turbine wake model Fuga, Report E-0046, DTU Wind Energy., 2014. /***

P10L15: While wind and nautical terms can easily be construed to have a connection, not everyone is familiar with "starboard".

***/ Sorry, being offshore tempt us to use nautical terms.
"Starboard" replaced by "right"
"Port" replaced by "left" /***

Figure 1 center: Should that really be called Uncertainty, or rather something like Deviation?

***/ We have updated Fig. 1. (see below). We are calculating an uncertainty value for each 10-min SCADA measurement. And the input for this calculation comes from the SCADA power, the $N$ and the LUTs. This uncertainty is a variable threshold for the model. Only if the deviation between the ratio $\pi$ and the ratio $\mu$ is

higher than the uncertainty, then underperformance is highlighted. For this reason we will keep the wording "uncertainty" in Fig.1. /***

[Figure]

/***

**Part 2**

[revised manuscript text omitted]

**Kommentar [MN8]: RC1: 5**. Page 1 lines 27-28 "Work on: : : of 2016". You don't need this reference and does not help the paper so remove it

**Kommentar [MN9]: RC1: 6.** Page 1 lines 28- Page 2 line 1 "For most turbines: : :. wake effects" You make it sound
as it was only a problem for offshore turbines and it is not so replace by e.g. "For most
turbines in a typical wind farm, verification of the performance by comparison with the
power curve is not suitable: : :" then ": : : maintenance of a met mast is very expensive
particularly offshore."

**Kommentar [NM10]:** RC3: Page 2 Line 1: Offshore met masts are very expensive. Are people really putting up met masts offshore to verify the turbine performance? Or onshore?

**Kommentar [MN11]: RC1:** 7. Page 2 line 3 Replace "accounts" by "account"

**Kommentar [MN12]: RC1:** 8. Page 2 line 6 "Incorrect parameter settings" you mean "turbine parameter"?

**Kommentar [MN13]: RC1:** 9. Page 2 line 9 "turbine has a limited power output which has been externally applied"
You want to refer to the limit but with the use of the "which" you mean the limited power
output but surely this is not what is externally applied because one cannot apply a
limited power output: : : that is a consequence of limiting something else.

**Kommentar [MN14]: RC1:** 10. Page 2 lines 11-16 this is a very weird paragraph. "Upwind turbines influencing the free flow for downwind turbines". This is a weird sentence because the flow is not
free for the downwind turbines. Why not just removing the "free" word. Then it is also
weird because you have ": : :, wake effects," so with this construction you say the wake

An international  working group (IEC TC88 WG6, 2005) was trying to come up with a standard for wind farm power performance testing The proposed  uses  met mast to establish a measured wind farm power curve matrix. This two dimensional measured power matrix (Wind direction, Wind speed) is compared against a modelled power matrix taking wake effects into account (Mellinghoff, 2007)(Carvalho and Guedes, 2009). The standard could not be established.

 (Mittelmeier et al. (2013) presented a  where not the absolute values between model and measurement are compared, but the relations between an observed turbine and all other turbines in the farm. In this way, the uncertainty of the measurement chain could be reduced. The model is also based on pre-calculated power matrices which we call from now on "lookup-tables" (LUTs). Different wake models or even combinations of wake model results can be used to provide results for these LUTs. But the model relies on measurements from a met mast which is often not available.~~to compare expected power results generated from complex wake models (pre-calculated and stored in matrices) with the actual wind turbine power output to detect underperformance in multiple wake situations. To lower uncertainties the method is based on ratios between the observed turbine and all reference turbines in the wind farm. This method relies on measurements from a met mast to determine the environmental conditions. Especially for offshore sites, a met mast is very expensive and therefore often not available.~~ Furthermore, with increasing size of wind farms, the assumptions of one measurement position being representative for the whole offshore wind farm is not valid (Dörenkämper, 2015). Further investigations are necessary to obtain a reliable and automated method to detect underperformance at individual turbines in a wind farm.

The purpose of this paper is to present the results of extending the wind farm performance monitoring method of Mittelmeier et al.  2013) by using SCADA instead of met mast data. A new combination of methods to obtain representative environmental condition and further optimisation potential for wake models fine-tuned by SCADA data is presented and an estimation of the uncertainty of these methods is given.

In Section 2 the general  by Mittelmeier at al. 2013) is recalled. A new approach to generate a virtual met mast from SCADA data is explained in detail in Section 2.1. The wake model optimisations are described in Section 2.2. A closer look at the uncertainties of the method especially in relation to the establishment of a virtual met mast is discussed in Section 2.3. In Sections 3, 4 and 5 results for a demonstration case are presented, followed by a detailed discussion and the final conclusions.

**Kommentar [MN15]: RC1:** 11. Page 2 line 18 ": :which proposed" so the standard stopped at some point proposing this?

**Kommentar [MN16]: RC1:** 12. Page 2 line 19-20 ": : :,could not be established," So it is not a standard, it is a working group trying to come up with a standard. Also delete the part ": :and the support: : : crumbled." It is not scientific knowledge

**Kommentar [MN17]: RC1:** 14. Page 2 lines 23-27 "Mittelmeier et al. (2013) presented: : :. environmental conditions." I am not sure this is a new method. In many other studies, authors used wake-model-based LUTs to estimate the efficiency of the wind farm. So the authors need to explicitly say what exactly is new.

**Kommentar [MN18]: RC1:** 13. From line 22 in page 2 onwards you talk about "matrices" but what you mean is "look-up-tables (LUT)" Use that term. There is an unnecessary comma in line 22 after "method". Also in that line you talk about detection of "curtailment". Perhaps your ...

**Kommentar [NM19]:** RC3: P2L25: To lower uncertainties, ... ??

**Kommentar [MN20]: RC1:** 15. Page 2 lines 27-28 "Especially: : : available." Already mentioned so remove it

**Kommentar [MN21]: RC1:** 16. Page 2 lines 31-33 Replace "this" by "the", add "of" after "method" and use "Mittelmeier et al. (2013)" instead of "(Mittelmeier et al., 2013). Replace "condition" by ...

**Kommentar [MN22]: RC1:** 17. Page 3 line 1-2 "Hence the presented: : : LiDAR". Based on what you have already mentioned one can inferred what is written here so it is not necessary

**Kommentar [MN23]: RC1:** 18. Page 3 line 4 it should be change to ": : :.of the method by Mittelmeier et al. (2013) is recalled".

**2. Methods**

To detect underperformance of a wind turbine, we estimate the expected turbine power  $\pi$  (predicted power ratio) between the observed turbine and a reference turbine with a wake model for the actual condition and compare its result with the actual measured power  $\mu$ . A deviation between $\pi$ and $\mu$  higher than a certain threshold indicates

5 underperformance.

10
 The performance monitoring model (Fig. 1) is based on two dimensional LUTs. The user can choose any wake model or even a combination of different model results to provide power output $P_{\pi\,i,j}$ values for different

15 wind speed bin $i$ and wind direction bin $j$. The predicted power output $P_\pi$ is derived from the  LUTs with linear interpolation knowing the measured wind speed and wind direction.

Information about the turbulence intensity, pressure, temperature and humidity from additional devices could be used to increase the dimensions of the power matrix and may add accuracy. As we are focusing on a monitoring method that

20 uses only SCADA data, we will discuss and demonstrate one way to extract an useful wind speed and wind direction for this  monitoring method in Section 2.1.

Commonly used power measurements are averages  over 10-min periods . Due to the fact that there is a high scatter on power measurements for the same wind speed and wind direction bin, averaging a number $N$  of 10-min

25  samples is necessary until the power value converges to a satisfactory degree. The power matrix   $N$ are derived in a pre-process as shown in Figure 1, which gives an overview on the whole performance monitoring process.

The power of the wind turbine under observation $P_{ob}$ is  divided by the power of a reference wind turbine $P_{ref}$.

30 This leads to a normalized power curve with much lower slope in a wide range of partial load (See Figure 2 ) and therefore decreases sensitivity on wind speed measurement uncertainty. We define

$$\mu = \frac{1}{N}\sum_{n=1}^{N}\frac{P_{\mu ob_n}}{P_{\mu ref_n}} \text{ and,} \tag{1}$$

$$\pi = \frac{1}{N}\sum_{n=1}^{N}\frac{P_{\pi ob_n}}{P_{\pi ref_n}}, \tag{2}$$

where $P_{\mu ob}$ and $P_{\pi ob}$ are the measured and predicted power of the observed turbine. $P_{\mu ref}$ and $P_{\pi ref}$ are the measured and predicted power of the reference turbine.

The underperformance indicator is defined as

$$\eta_{ob,ref} = 100 \% \cdot \left(1 - \frac{\pi}{\mu}\right). \tag{3}$$

 Having the measured power correlation in the nominator increases the sensitivity. The underperformance interval range of the indicator is in this way between $[0, -\infty[$. Non-operating turbine values have to be filtered out.

If $\eta_{ob,ref}$ is larger than the uncertainty (Section 2.3), underperformance has been detected. This correlation is repeated for each combination of turbines which leads to $n \cdot (n - 1)$ results (n = number of turbines in the farm). This adds further confidence to the detection, because an underperforming turbine will meet the criteria several times.

**2.1 Determination of environmental conditions**

**2.1.1 Wind direction**

The first step is to derive a wind direction $\vartheta$ for each 10-min interval. For our monitoring model we are using the absolute wind direction signal from each turbine which is defined as

$$\vartheta = nacelle\ position + wind\ vane\ position. \tag{4}$$

The nacelle position is the angle between the rotor axis and a marking for true north. This marking is calibrated as part of the commissioning. But often this signal is not maintained well during operation, because it has no effect on turbine performance. This causes the necessity to apply an offset correction to this signal before using it for reanalysis purposes. The wind vane position indicates the angle of the flow to the rotor axis. It directly provides a value for the yaw error. The turbine controller uses this signal to control the yaw activity.

**Kommentar [MN36]: RC1:** 29. Page 4 line 12 Replace "can be described with Eq. (3)" by "is defined as"

**Kommentar [MN37]: RC1:** 30. Page 4 lines 16-17 "where neta_ob,ref: : : (ob) turbine" You already defined everything so there is no need for this

**Kommentar [NM38]:** RC2: In section 2.1.1 the wind direction is derived, but without any reference to how this is done. The wind direction measured on the nacelle is only used for yaw control, where the strategy is the keep the rotor aligned with the wind direction to minimize the yaw-misalignment. This signal can also identify a "forced" yaw misalignment used to determine the "wake drift"? The optimal readings from this instrument is 0, and will not reveal anything about the actual flow direction, which only can be identified from the wind turbine yaw position. The wind turbine yaw position not used by the controller, only when wind farm has sector management (proposed but never seen). The yaw position is usually not calibrated or has a wrong offset, which need to be identified.

**Kommentar [NM39]:** RC3: 2.1.1: I wonder why you do not use the same correction method on all the wind farms wind vanes? It just requires SCADA data and some computer power, so it would not be too difficult. Also, when you're calculating a mean wind direction for the whole farm, aren't you relying on a smaller size farm far away from the coast? For example, if you have a location like Anholt, then you have wind and direction gradients due to the proximity of the land across the wind farm. How would that influence your method and its accuracy?

Within the Pre-Process (Fig.1) of the monitoring model we estimate the north marking offset for one turbine by checking the location of the maximum wake deficit with respect to the true north. Then we compare the average wind direction between corrected turbines and neighbouring turbines to estimate the remaining offset for all turbines.  After applying this offset correction, the wind direction from all wind vanes are averaged in the complex  plane to account for the wind direction discontinuity at the beginning/end of the  value range, after removing outliers,  outside $\pm1.5 \cdot$ IQR (interquartile range).

Kommentar [MN40]: RC1: 31. Page 4 lines 26-29 So why do you have to use all the wind vanes (this is what is read from the text)? They could all have a different misalignment and so you will need to analyze each of them (in terms of wake deficits) if you want to use all of them.

Kommentar [MN41]: RC1: 32. Page 4 line 29 "complex area" what do you mean by complex area?

Kommentar [MN42]: RC1: 33. Page 5 line 1 "of the scale" what do you mean by scale?

Kommentar [MN43]: RC1: 34. Page 5 line 2 "+-1.5IQR" be explicit. If the outliers removed are those outside the range +-1.5IQR then say so

**2.1.2 Wind speed**

Having determined an averaged wind direction we are now able to derive the averaged free flow wind speed. For this task we use the nacelle anemometry but only from wind turbines that are not affected by upwind turbines. To determine whether a turbine is affected by an upwind turbine or not we use the specification for power curve measurements from the international standard (IEC 61400-12-1, 2005). Each turbine location is checked against all other turbine locations according to the averaged wind direction. This is done within a Cartesian coordinate system were $x$ represents the easting and $y$ being the northing (See Figure 3). The wind turbine of interest $WT_i$ is located at the position $(x, y)$ and the turbine wake is from the turbine $WT_0$ at location $(x_0, y_0)$.

$$\alpha = 1.3 \cdot \arctan\left(2.5 \cdot \frac{D_n}{L_n} + 0.15\right) + 10,$$

(5)

 is proposed by (IEC 61400-12-1, (2005) and  defines the width of the disturbed sector  in degrees seen by the downwind turbine (the constants have the dimension of degree).  $D_n$ is the rotor diameter of the upwind turbine and $L_n$ the distance between the two turbines  defined by Eq. (5).

Kommentar [MN44]: RC1: 35. In Eq. 4 you have constants without units. If alpha is in degrees all these constants have the units of degrees and you need to state that

$$d_x = |x - x_0|,$$
$$d_y = |y - y_0|,$$
$$L_n = \sqrt{d_x^2 + d_y^2},$$

(6)

$$\beta = \begin{cases} \frac{\pi}{2} - \arctan\left(\frac{d_y}{d_x}\right) & x_0 > x \text{ and } y_0 > y \\ \frac{\pi}{2} + \arctan\left(\frac{d_y}{d_x}\right) & x_0 > x \text{ and } y_0 \leq y \\ 0 & x_0 = x \text{ and } y_0 > y \\ \pi & x_0 = x \text{ and } y_0 \leq y \\ \frac{3}{2}\pi - \arctan\left(\frac{d_y}{d_x}\right) & x_0 < x \text{ and } y_0 \leq y \\ \frac{3}{2}\pi + \arctan\left(\frac{d_y}{d_x}\right) & x_0 < x \text{ and } y_0 > y \end{cases}, \tag{6}$$

 With $\beta$, being the angle between the wake inducing turbine and the northing and the wind direction $\vartheta$ the turbine wake indicator $\gamma$ can be described as:

$$\quad \gamma = \begin{cases} |\beta + 360 - \vartheta| - \frac{\alpha}{2} & 0 < \beta < 90 \text{ and } 270 < \vartheta < 360 \\ |\beta - 360 - \vartheta| - \frac{\alpha}{2} & 270 < \beta < 360 \text{ and } 0 \leq \vartheta < 90 \\ |\beta - \vartheta| - \frac{\alpha}{2} & \text{else} \end{cases}, \tag{7}$$

The wind turbine of interest $WT_i$ is categorized as waked turbine for $\gamma < 0$. The wind speed for the virtual met mast is therefore the average of the subset of the nacelle anemometer signals from all wind turbines with $\gamma > 0$.

**2.2 The wake model**

10   The wake model is a key factor in our performance monitoring method. Several benchmark tests have been published with a large variety of different models (Gaumond et al., 2012), (Réthoré et al., 2013) and (Steinfeld et al., 2015). And research is still ongoing to further improve prediction accuracy of such models.

In Figure 1 we highlight that the wake model and its tuning is part of the pre-process. The performance monitoring method
15   itself is based on linear interpolation from the LUTs only. In (Mittelmeier et al., 2015), we have identified three key parameters for the tuning of the wake model (stability, wind direction uncertainty and wake drift). Figure 2 gives an example of how the different key parameters change the wake model results. The left plot visualises the active power of a turbine in wake normalised with a free flow condition in 6.3 D distance. The 0° on the *x*-axis locates the full wake situation according to the simulation. The right plot is a representation of the same data as normalised power curve with wind speed on the *x*-axis normalised with the wind speed, when wake effects fade away due to pitching activities of the upwind turbine.

In the first step, the wake model needs to be set up with the right atmospheric stability parameters. An increasing stability will cause higher wake losses and therefore shift the wake plot vertically down (from red rhombus to black triangles).

Kommentar [MN45]: RC1: 36. Eq. 6 is not needed

Kommentar [MN46]: RC1: 37. Page 6 line 1 "the north inconsistency need different conditions" Yes obviously

The next two steps are applied on the wake model results which need to be calculated for a directional resolution of 0.5° and for each wind speed bin of 1 m/s. This resolution was proposed by Gaumond et al.(Gaumond et al., 2014) for his method to account for measurement uncertainties related to the wind direction which is the second key parameter in our tuning process. In his paper, three main sources of uncertainty are mentioned: The yaw misalignment of the reference turbine, the spatial variability of the wind direction within the wind farm and the variability of wind direction within the averaging of a 10 min interval. This causes a higher scatter in the data and leads to averaging effects that are not modelled in the simulation. In a post process each wind direction is averaged with weighted neighbouring results. A Gaussian distribution with a standard deviation $\sigma_a$ has been proposed as a weighting function. The effect of this step is visualised in Figure 2 (red rhombus are without and orange points are with $\sigma_a$ weighted averaging). In (Mittelmeier et al., (2015), we could show, that for the prevailing conditions at Ormonde wind farm $\sigma_a$ is a function of the wind speed, decreasing with higher wind speeds.

Looking at the full wind rose for an AEP estimation, the Gaussian averaging has little no impact on the result (Gaumond et al., 2014). But the smaller the wind direction bin size, the larger the prediction error made by the wake model. Hence it is crucial for our monitoring method to increase accuracy for smaller wind direction bin sizes which will decrease the uncertainty of the method.

The third tuning parameter is applying a simple offset on the wind direction of the LUTs to account for a drift of the wake. We call this phenomena from here on "wake drift". (Fleming et al., (2013) has studied the effects of active wake control and in his baseline simulation (no yaw error) a small wake drift to the right can be observed when looking downwind. In the LES study of (Vollmer et al., (2016) the wake drift increases from neutral to stable conditions also for 0° yaw angle. (Gebraad, (2014, p86) gives an explanation for the observations from the simulations by Fleming et al. (2013). The flow reacting on the rotation of the rotor causes the wake to rotate counter clockwise (looking downstream). Higher wind speeds from the upper layer are transported downwards (on the left side) and lower wind speeds from the lower layer are pushed upward on the right side of the wake. As a result the velocity deficit at the right part of the wake increases, so the wake deflects to the right. (Marathe et al., (2015) could show in their field measurement campaign with a dual-doppler radar the wake drifting to the right, as expected by the theory. But in the far wake they registered a movement to the left. The authors state the hypothesis that this contradicting phenomenon may be caused by atmospheric streaks. In an offshore field experiment by (Beck et al., (2015) further evidence is provided that wakes are moving out of the centre line. This wake drift is currently not modelled in Fuga and therefore applied in a further step of the pre-process (Fig. 1).

**Kommentar [MN47]:** RC1: 38. Page 7 lines 1-2 "In Mittelmeier et al. (2015): : : wind speeds". Well that depends on the stability conditions. This will be true if compare unstable conditions with low wind speeds and high sigmas with neutral conditions with lower sigmas and higher wind speeds. But stable conditions will be in the low wind speed range with lower sigmas compared to neutral

**Kommentar [MN48]:** RC1: 39. Page 7 line 3 "no impact" you mean "little impact"

**Kommentar [MN49]:** RC3: P6L15: You might want to explain what you mean by "wake drift".

**Kommentar [NM50]:** RC2: The description of the method seems to be adequate, but the "wake drift" in section 2.2 is not well defined, I assume this refers to periods with active wake control, which I do not expect has been implemented yet?

**2.3 Uncertainties and underperformance criteria**

[revised manuscript text omitted]

**3 Results and Demonstration**

The objective of this paper was to present a developed method, using only SCADA data and pre-calculated numerical wake model results to detect underperformance at wind turbines in waked conditions within the wind farm. We have chosen the Ormonde wind farm to demonstrate the new method. The 30 turbines have a rated power of 5 MW and are owned by Vattenfall. The wind farm is located in the Irish Sea 10 km west of the Isle of Walney.

The farm layout displayed in Fig. 5 is structured in a regular array which allows comparison of several  wake situations. The closest turbine spacing is in the range of 4.1 D to 4.3 D along the four rows

Kommentar [MN54]: RC1: 42. Page 8 line 19 "is around 7%" based on what?

Kommentar [MN55]: RC1: 43. Page 9 line 3 Figure 5: please show a proper layout with north orientation and scales

orientated from north west to south east. We have selected a more frequent wind direction from south-south west where multiple columns of four turbines are aligned with a distance ranging from 6.3 D to 6.5 D . To simplify the demonstration of underperformance detection we focused on single wake, double wake and triple wake conditions behind turbine  OR26 for a south-south westerly wind direction and a sector of 30° around the full wake situation. Two years of 10-min SCADA data were used to set up the performance monitoring model.

**3.1 Environmental condition of demonstration wind farm**

**3.1.1 Wind direction**

The wind direction is supposed to be representative for the wind turbines in the monitoring model. In our example, we have averaged up to 30 corrected wind direction signals for each 10-min interval. The variation among the individual signals provides an uncertainty estimate for this artificial wind direction. In Fig. 6, a histogram of the full data set of two years with each count being the difference between a single vane measurement and the corresponding mean wind direction for the averaged period is visualized. This variation can nicely be described by a Gaussian distribution with standard deviation of 3.6 °. This value is used for the uncertainty of the wind direction Table 1 is referring to.

~~The quality of the derived wind direction is visualized by plotting a histogram (Figure 6) for the full data set of two years with each count being the differences between a single wind vane measurement and the corresponding mean wind direction for the averaged period. The deviation of the single wind vanes from the averaged wind direction is nicely described by a Gaussian distribution with standard deviation of 3.1°. This value is usedfor the uncertainty of the wind direction Table 1 is referring to.~~

**3.1.2 Wind speed**

Figure 7 demonstrates the quality of the virtual met mast derived with the methodologies described in Section 2.1.1 and Section 2.1.2. The average wind speed of all nacelle anemometers has been normalised by the averaged nacelle anemometer wind speed of the wake free subset. The full data is binned into 2° and plotted against the averaged wind direction. The errors bars indicate the experimental standard deviation of the mean (JCGM, 2008). We obtain a quite good agreement with the Fuga model which has been used with a Gaussian averaging of standard deviation $\sigma_a = 4$ °. So far there is no instruction available on how to determine this standard deviation which should take wind direction uncertainty into account (Gaumond et al., 2014). We have chosen this value, because of a quite nice fit with the SCADA data. A linear regression between the wind speed of the standard model results (red dashed line) and the SCADA measurements equals $R^2 = 0.96$. The improved model (green solid line) gives an $R^2 = 0.98$.

When considering the demonstration sector of 30° around the full wake alignment behind wind turbine 26, the free flow wind speed can also be described by a Gaussian distribution (Figure 8) with a standard deviation of 0.46 m/s.

This information is important for the investigation of the uncertainties Table 1 is referring to.

**3.2 Wake model**

5 For the demonstration of the described method, we used the Fuga wake model which uses linearized Reynolds Averaged Navier Stokes equations developed by Ott et al. (2011). With the second version of the software new features were added (Ott and Nielsen, 2014) to account for different atmospheric stabilities and for wind direction uncertainties. The results for this paper have been produced with Fuga version 2.8.4.1. We have chosen this wake model for two reasons: Firstly, there is already a confident number of validations with measurements published (Gaumond et al., 2012), (Mortensen et al., 2013)

10 and (Steinfeld et al., 2015) and secondly, the Gaussian averaging feature described by Gaumond et al. (Gaumond et al., 2014) is already implemented.

To get a more reliable monitoring model we need to calibrate the wake model settings and compare several different calculation results with measured SCADA data based on the established virtual met mast. The wake model is supposed to

15 provide a two dimensional LUT (wind direction, wind speed) for each turbine. Further dimensions such as stability may improve the accuracy, but research and validation for these models are still ongoing. Therefore our calibrated model has to be representative for the average annual conditions. Two full years of SCADA data have been used for this task. We identified three steps to obtain a better match between the power modelled by the wake model Fuga and the measurements. Firstly, the standard deviation $\sigma_a$ to account for the wind direction uncertainty was found to be decreasing

20 with increasing wind speed. Secondly, the Fuga parameter to model the effect of atmospheric stability $\zeta_0 = 2.72E - 7$ was set to more stable conditions and thirdly, the centre of the wake was found to be  drifting towards the right side when traveling downwind. Approximately 2.5° in the single wake and an additional 1° is added with every turbine adding an additional wake to the flow. This results in a total offset of 4.5° for the triple wake referenced to the artificial wind direction from the virtual met mast (Section 2.1.1). One possible explanation for this behaviour is the

25 fact, that the upwards moving blade diverts the flow with higher wind speeds downwards to regions with lower wind speeds and the downwards moving blade causes the opposite. This results in a higher wind speed on the port side than on the right side of the wake and leads to a drift of the wake centre. A second explanation can be derived from the Coriolis force, which leads to an increased force to  right (northern hemisphere) on accelerating air particles. We cannot fully rule out the possibility of an unwanted yaw misalignment as the uncertainties within this process of aligning the turbine

30 lies within 3° (IEC 61400-12-2, 2013). But a single wake drift of 2.5° is also within the simulation results for 0° yaw misalignment at stable conditions (Vollmer et al., 2016).

**Kommentar [MN64]: RC1:** 50. Page 9 The wake model seems to have a systematic bias when compared to the measurements (clearly seen at the highest deficits). Why? Is the average wind direction perhaps wrong? Perhaps you should average the wind direction of the turbines in the rows where the flow is not disturbed if you want to compare it with the wake model

**Kommentar [MN65]: RC1:** 51. Page 10 line 1 The wake model should be presented before the results in 3.1.2!

**Kommentar [NM66]: RC3:** P10L5: I would drop the brackets around the version number of Fuga.

**Kommentar [MN67]: RC1:** 52. Page 10 line 11 How much data you use for the calibration period?

**Kommentar [NM68]: RC3:** P10L14: I don't think I've seen zeta_naught introduced before?

**Kommentar [MN69]: RC1:** 53. Page 10 starboard and port terms are terms conventionally used in wake studies?

**Kommentar [NM70]: RC3:** P10L15-22: This is quite interesting. Has this behaviour been observed anywhere else? Another explanation could be that the overall wind flow is skewed at the Ormonde location, which judging by the map is not impossible, seeing that the wind farm is wedged between land and a larger offshore wind farm. Or do I understand this ...

**Kommentar [MN71]: RC1:** 54. Page 10 line 17 what do you mean by "global"

**Kommentar [NM72]: RC3:** While wind and nautical terms can easily be construed to have a connection, ...

**Kommentar [MN73]: RC1:** 55. Page 10 lines 16-22 If this phenomenon occurs in 1 single wake the a very plausible ...

Figure 5 shows the location and layout of the demonstration wind farm.  The  column of turbines behind turbine 26 has been selected for the validation of the wake model settings. The benchmark are simulations for neutral conditions with none of the post processing's mentioned in section 2.2 to take wind direction uncertainty, atmospheric stability and wake  drifts into account. Figure 9 demonstrates the improvement of model prediction and its capabilities for single wake, double wake and triple wake situation. The left column visualises wake deficit plots where the power has been normalized with the free flow turbine, as function of the wind direction, centred to the full wake. The data is filtered for wind speed of 8±1 m/s. The right column are normalized wake power curves. The power, normalized with free flow power, is shown as function of the wind speed, normalized with wind speed at rated power for the  turbine in the wake. Thes data has been filtered for a wind direction sector of 5°. The optimised simulation results with the green diamonds follows the SCADA data with the black dots much closer than the benchmark case marked as red triangles. The error bars indicate one standard deviation of the measured SCADA data at each bin. The three fine-tuning steps decreased the power prediction error in a full wake with ±5° sector width from 7% to 1.5%(Mittelmeier et al., 2015) for the presented case.

Having now an optimised wake model, the first two steps of the pre-process (Figure 1) are accomplished and the matrices for the "predicted power" can be established. In the next Section, the detection of underperformance will be demonstrated with  two test cases.

**3.3 Demonstration Case**

Two years of SCADA data have been contaminated with two different error types. The first manipulation simulates a degradation of 8 % of its power production for which the original data set that has been used to calibrate the model, is multiplied by 0.92. According to the findings in Section 2.3 a degradation of 8 % is just high enough to distinguish from the uncertainties of a turbine in triple wake. The second test case is a simple power curve curtailment at 60 % rated power.

In Figure 10 the normalized power as function of the normalized wind speed is shown in a scatterplot. The coloured points in green represent correct turbine performance (P_optimal). The yellow dots (P_degraded) describe the degradation and the red dots (P_curtailed) are the data with the curtailment. Measurements from above rated wind speed are removed to concentrate on the part of the power curve where underperformance is more difficult to detect.  Below 5m/s we have realized a strong increase in wind direction variation among the turbines compared to the artificial wind direction from the virtual met mast. This variation increases the uncertainty of the model and therefore it is filtered out.

Kommentar [MN74]: RC1: 56. Page 10 line 32 "this data has been filtered for a wind direction sector of 5 deg" Change to "These data have been.."

Kommentar [MN75]: RC1: 57. Page 11 line 2 You mean for prediction of what? AEP? Wind speed? A particular case?

Kommentar [MN76]: RC1: 58. Page 11 line 5 Replace "has been" by "will be"

Kommentar [MN77]: RC1: 59. Page 11 line 6 what do you mean by "real data"? so before the data was not real?

Kommentar [MN78]: RC1: 60. Page 11 line 9 degradation of 8% in terms of what?

Kommentar [MN79]: RC1: 61. Page 11 line 13 Replace "in displayed" by "is shown"

Kommentar [MN80]: RC1: 62. Page 11 line 18 Why "Therefore"?

[revised manuscript text omitted]

---

## Author Response (AR2)

Dear Referee RC1,

We would like to thank you very much for all your effort you have put into the review of our paper. You will find below our answers to your comments marked with ***/ response /***.

RC1 Minor comments:

1. Page 1 line 9: do not define acronyms in the abstract

***/ Changed /***

2. Page 1 lines 12-13: "averaging several measurement devices" One cannot do this but averaging measures (of the same type) performed by devices

***/ Changed /***

3. Page 1 lines 20-21: "status of not being restricted in power production" Even if such a phrase is stated in the standard please try to use a better English form to say so

***/ New sentence:

"However within this standard the "Full Performance" category requires only a turbine status signal which confirms a power production without any restrictions but there is no verification of the quality of the power performance."

/***

4. Page 2 line 2: delete "Using". Also it looks like a "to find" is missing perhaps after "requires" and replace "leads" by "can lead"

***/ All changes applied:

"This approach still requires to find a wake free sector and can lead to an increase in uncertainties"

/***

5. Page 2 line 3: Once again this should be written as "in uncertainties (Albers et al., 1999, IEC 61400-12-2. 2013).

***/ Changed/***

6. Page 2 line 16 "high uncertainties" of what?

***/ New sentence:

"Due to high uncertainties in flow and wake modelling this method is only proposed as a first general check."

/***

7. Page 2 line 19 you mean "affecting" instead of "effecting"?

***/ Changed/***

8. Page 2 line 25 should read "(Mellinghoff, 2007, Carvalho and Guedes, 2009)."

***/ Changed/***

9. Page 2 line 26 it seems like you need a "because…"

***/ In our first version we mentioned that there was not sufficient support anymore to accomplish the Standard. In your comment NR 16 you commented that this is not scientific knowledge, so we decided to remove it. /***

10. Page 2 lines 28-29 should read "… a new method that uses relations between an observed turbine and all other turbines in the farm instead of the absolute values between model and measurements"

***/ Changed/***

11. Page 2 line 30 replace "model is also based on" by "method uses"

***/ Changed/***

12. Page 2 line 32 replace "model" by "method"

***/ Changed/***

13. Page 3 line 7 "of these methods" you are showing only one method… that uses an expression for the uncertainty and a wake model

***/ Changed/***

14. Page 4 line 16 delete "Having the measured… sensitivity"

***/ Changed/***

15. Page 4 line 20 not the same n symbol in the three instances used

***/ Changed/***

16. Page 4 line 21 replace "criteria" by "uncertainty criterion"

***/ Changed/***

17. Page 5 line 1 replace "These causes the necessity" by "We need" and delete "for reanalysis purposes"

***/ Changed/***

18. Equation 5. You are bringing the equations in a weird way. Equations are part of the text so in this particular case you finished a sentence "(xo,yo)." and suddenly you have the equation.

***/ New sentence:

"The width of the disturbed sector in degrees seen by the downwind turbine is defined as

$$\alpha = 1.3 \arctan\left(2.5\,\frac{D}{L} + 0.15\right) + 10 \qquad (1)$$

where the constants have the dimension of degree (IEC 61400-12-1, 2005)"

/***

19. Page 5 line 21: All the constants in equation 5 have the dimension of degree?

***/ New  syntax:

"The width of the disturbed sector in degrees seen by the downwind turbine is defined according to the IEC 61400-12-1 (2005) as

$$\alpha = 1.3 \arctan\left(2.5\,\frac{D_n}{L_n} + 0.15\right) + 10° \ . \qquad (2)$$

/***

20. Page 5 line 28: Add a comma after "wind direction theta"

***/ Changed/***

21. In section 2.1.2. you describe the method to find out the wake affected turbines. Why not using the wake model for such analysis?

***/ You are right, another way of selecting wake free turbines would be to compare the wind speed of the wake model. We would need to define a tolerance band and select for each wind direction the turbines at which location the wind speed is within the tolerance band of the simulated wind speed. We have not compared both methods yet because the proposed method is working fast and reliable. But we will consider a comparison in the future. Thank you for this suggestion. /***

22. Page 6 line 7 Again bad referencing

***/ Changed/***

23. Page 6 line 11 Change the text after the stop to "In Mittelmeier et al. (2015) three key parameters for the tuning of the wake model are identified"

***/ Changed/***

24. Page 6 line 13 replace "visualizes" by "shows"

***/ Changed/***

25. Page 6 line 14 replace "in 6.3 D" by "at 6.3 Dn" I guess it is Dn not D only

***/ Changed, We have decided to delete the suffix and only use $L$ for the distance between the turbines and $D$ as rotor diameter of the upwind turbine/***

26. Page 6 line 16 delete the comma after speed

***/ Changed/***

27. Page 6 line 21 Again a problem with the reference… should read "…proposed by Gaumond et al. (2014) to account for…"

***/ Changed/***

28. Page 7 line 6 replace "phenomena" by "phenomenon" and delete "has"

***/ Changed/***

29. Page 7 line 15 delete "as expected by the theory"

***/ Changed/***

30. Page 7 line 17 you speak about Fuga suddenly

***/ Sentence deleted. We have mentioned before, that this effect is modelled in the pre-process wake model tuning part. So this sentence is not needed here. Fuga is introduced in Section 3.2.

/***

31. Equation 8: what is K?

***/ $K$ is the total number of uncertainty components in the measurement chain. Definition has been added/***

32. Page 7 line 31-Page 8 line 1 reduce/rephrase that sentence because it is too long and not clear

***/ sentence rephrased:

Table 1 shows the uncertainty components of the predicted power $P_\pi$ and provides the sensitivity factors. $P_{\pi\ i,j}$ is the power value in the matrix referring to the wind speed bin $i$ and the wind direction bin $j$. $V_{i,j}$ is the wind speed and $\vartheta_{i,j}$ the wind direction of the element."

/***

33. Page 8 line 7: replace "include" by "includes"

***/ Changed/***

34. Equation 9 make it in one line and delete the dot before the square root because it is not a dot product

***/ Changed:

$$u(\mu) = u\left(\frac{P_{\mu ob}}{P_{\mu ref}}\right) = \frac{P_{\mu ref}}{P_{\mu ob}}\sqrt{\left(\frac{u_c(P_{\mu ob})}{P_{\mu ob}}\right)^2 + \left(\frac{u_c(P_{\mu ref})}{P_{\mu ref}}\right)^2}. \qquad (3)$$

/***

35. Page 8 line 17: delete "Equation 9 explains…. (Bell, 2001)"

***/ deleted/***

36. Page 8 line 20: should read "according to Ku (1966)"

***/changed/***

37. Page 9 delete sentence between lines 8-9

***/ deleted/***

38. Page 9 line 14 replace "have selected" by "select"

***/changed/***

39. Page 9 line 21 delete first sentence

***/ deleted/***
40. Page 9 last line replace "has been" by "is"

***/ replaced/***

41. Page 10 line 3 delete "a Gaussian averaging of standard deviation"

***/ deleted/***

42. Page 10 line 5 add "The coefficient of determination of " before "A linear regression"

***/ added/***
43. Page 10 lines 18-19 problems with the references and I am tired so it is up to you

***/changed/***

44. Page 10 line 21 replace "model" by "method"

***/changed/***

45. Page 10 from ~line 22 to Page 11 ~line 2 there are many sentences that were used before such as the sentence starting with "the wake model is supposed to" or "Therefore, our calibrated model has to be representative" and the wake drift explanation

***/ You are right, there is some sort of repetition. In Section 2.2 we describe the tuning in a generalized way and in Section 3.2 we explain how the general ideas are executed with the selected wake model Fuga and which results we obtain from the three steps. We hope that this similar structure helps the reader.

/***
46. Page 10 line 25 replace "have been" by "are"

***/changed/***

47. Page 10 line 26 replace "identified" by "identify"

***/changed/***

48. Page 10 line 27 replace "to be decreasing" by "to decrease"

***/changed/***

49. Page 10 line 28 that is probably not the Fuga stability parameter but the dimensionless stability parameter… and the E should be e

***/ changed sentence:

"Secondly, the dimensionless parameter to model the effect of atmospheric stability $\zeta_0 = 2.72e - 7$ is set to more stable conditions"

/***
50. Page 10 line 29 "set to more stable conditions"-> how more? By how much?

***/ We have deleted "more". According to Fuga manual $\zeta_0 = 2.72e - 7$ is stable and it is not recommended to calculate with more stable cases than set by $\zeta_0 = 3e - 7$.

/***

51. Page 11 line 5 then you need to say what is the drift for the other stabilities?

***/ Sentence added:

"Whereas neutral and unstable conditions show no drift or even a very small drift in the opposite direction." /***

52. Page 11 line 8 "Figure 5… wind farm" you already showed and described this graph

***/ Sentence deleted /***

53. Page 11 line 12-15 sentences that were already stated

***/ We think it is helpful here in the context of the Section. /***

54. Page 11 line 15 replace "have been" by "are"

***/changed/***

55. Page 11 line 25 as above

***/changed/***

56. Page 12 line 1 replace "have realized" by "find"

***/changed/***

57. Page 12 line 2 "it is filtered out" you mean that you filter wind speeds below 5 m/s

***/ Sentence revised:

"This variation increases the uncertainty of the model and therefore wind speed below 5m/s are filtered." /***

58. Page 12 lines 8-11 already mentioned before

***/ Sentence deleted: "This is directly…into account." /***

59. Page 12 line 12 replace "appears" by "appear"

***/changed/***

60. Page 12 line 16 replace "has been" by "is"

***/changed/***

61. Page 14 line 2 tests instead of test

***/changed/***

62. Page 14 line 7 "were" instead of "have been"

***/changed/***

63. Page 14 line 10 reference?

***/ Sentence revised:

"Furthermore the uncertainties in performance level prediction could be reduced by normalization and cross-reference correlations." /***

64. Page 14 line 14 calibration no recalibration

***/changed/***

65. All graphs that use yellow should be changed and use other color

***/changed/***

66. Figs 6 and 7 should have [deg.] as units in the x-axis

***/changed/***

Dear Referee RC3,

we would like to thank you very much for once more reviewing our paper. Your comments helped a lot to improve the paper. Please find below our comments to your miner details marked with ***/ response/***

Minor details from RC3:

P3L9: Mittelmeier _e_t al._(2013)

***/ Changed /***

P7L6: Fleming et al. _have_ studied...

***/ Changed /***

Eq. 7, is that all bold?

***/ You are right, it was bold. Has been changed /***

P11L19: 1.5%_(Mittelmeier

***/ Changed /***

Finally, please go once more through the references and check whether all of them can be found easily based on the information you give, and check punctuation. Especially check the new Marathe et al, which is wrong on a number of counts.

***/ You are right, we have updated the references with the missing details /***

[revised manuscript text omitted]